# Complete attenuation of *Plasmodium falciparum* sporozoites by atovaquone–proguanil

Steffen Borrmann [1,2,3,10] ✉, Zita Sulyok [1,2,10], Katja Müller [4,10], Rolf Fendel [1,2,3,10], Mihály Sulyok[1,2,8], Johannes Friesen[4], Albert Lalremruata [1,2], Thaisa Lucas Sandri[1,2], The Trong Nguyen [1,2], Carlos Lamsfus Calle [1,2], Annette Knoblich[1,2], Stephanie Sefried[1,2], Javier Ibáñez [1,2], Freia-Raphaella Lorenz [1,2], Henri Lynn Heimann[1,2], David M Weller[1,2], Regina Steuder[1,2], Selorme Adukpo [1,2], Patricia Granados Bayon[1,2], Zsófia Molnár[1,2], Meral Esen[1,2], Wolfram Metzger[1,2], Eric R James[5], Adam Ruben[5], Yonas Abebe[5], Sumana Chakravarty[5], Anita Manoj[5], Natasha KC[6], Tooba Murshedkar[5], Julius C R Hafalla [7], Tamirat Gebru Woldearegai [1,2], Fiona O'Rourke[1,2], Jana Held[1,2], Pete Billingsley [5], B Kim Lee Sim [5], Thomas L Richie [5], Stephen L Hoffman[5,10], Peter G Kremsner[1,2,3,10], Kai Matuschewski [4,10] & Benjamin Mordmüller [1,2,3,9,10]

## Abstract

To develop a *Plasmodium falciparum* (Pf) vaccine that precludes replication inside the host for improved vaccine safety, we tested chemo-attenuation (CVac) of sporozoites (SPZ) with atovaquone–proguanil (AP). In mice, *P. berghei* sporozoites administered with AP invaded hepatocytes, arrested early, and induced robust protection, which correlated with parasite-specific effector-memory CD8+ T cell responses. In a clinical trial of PfSPZ-CVac (AP), in which three doses of $5.12 \times 10^4$ or $1.5 \times 10^5$ PfSPZ were administered by direct venous inoculation combined with oral single-dose AP (1000/400 mg), blood stage infections were fully prevented during immunisation. 2/8 and 2/10 of vaccinees, respectively, were protected when challenged with $3.2 \times 10^3$ PfSPZ 10 weeks later, inferior to PfSPZ-CVac (chloroquine/CQ) that allows in-host replication. Comparative analysis of responses to 228 Pf proteins revealed that protection with PfSPZ-CVac (CQ) was associated with antibodies to two liver-stage antigens (LISP2, LSA1) and a multi-stage antigen (PfMSP5), but not to the major surface protein PfCSP. The complete arrest of high numbers of Pf sporozoites by single-dose AP should allow a significant dose-frequency reduction of the current daily AP malaria chemoprophylaxis regimen.

**Keywords** *Plasmodium falciparum*; Malaria Vaccine; Atovaquone–proguanil; Controlled Human Malaria Infection; Chemo-attenuation
**Subject Categories** Immunology; Microbiology, Virology & Host Pathogen Interaction

See also: G M Rochfort Peters & J Baum

## Introduction

A global effort to fight malaria in all subtropical and tropical regions of the world has considerably reduced the disease burden. Progress, however, has recently even been reversed with an estimated 260 million cases and 597,000 deaths globally in 2023 (WHO, 2024). Further advances towards elimination are likely to be facilitated only by a highly effective vaccine that complements the existing conventional control measures such as vector control, long-lasting impregnated bednets and access to rapid diagnosis and chemotherapy (malERA Consultative Group on Vaccines, 2011).

Recently, an important milestone in malaria vaccine development was reached by the WHO recommendation of two monovalent paediatric vaccine formulations, namely, RTS,S/AS01 and R21/Matrix-M, for widespread use in endemic areas with moderate and high malaria transmission (Moorthy et al, 2024). Both vaccines, however, do not meet a previously defined vaccine efficacy (VE) of at least 75% for ≥2 years (Moorthy et al, 2013) in phase 3 trials. Both products are biosimilar, recombinant sub-unit vaccines targeting a single *Plasmodium falciparum* (Pf) sporozoite

[1]Institut für Tropenmedizin, Eberhard Karls University of Tübingen, Tübingen, Germany. [2]German Center for Infection Research (DZIF), partner site Tübingen, Tübingen, Germany. [3]Centre de Recherches Médicales de Lambaréné, Lambaréné, Gabon. [4]Department of Molecular Parasitology, Institute of Biology, Humboldt University, Berlin, Germany. [5]Sanaria Inc., Rockville, MD, USA. [6]Protein Potential LLC, Rockville, MD, USA. [7]Department of Immunology and Infection, Faculty of Infectious and Tropical Diseases, London School of Hygiene and Tropical Medicine, London, UK. [8]Present address: Institute of Pathology and Neuropathology, Eberhard Karls University of Tübingen, Tübingen, Germany. [9]Present address: Department of Medical Microbiology, Radboud University Medical Center, Nijmegen, The Netherlands. [10]These authors contributed equally: Steffen Borrmann, Zita Sulyok, Katja Müller, Rolf Fendel, Stephen L Hoffman, Peter G Kremsner, Kai Matuschewski, Benjamin Mordmüller. ✉E-mail: steffen.borrmann@uni-tuebingen.de

antigen, circumsporozoite protein (CSP), which is likely prone to the selection of vaccine escape variants (Neafsey et al, 2015).

Malaria is caused by single-cell eukaryotes of the genus *Plasmodium*. Morbidity and mortality result from rapid, asexual replication inside red blood cells. In the case of Pf, which accounts for nearly all malaria-related deaths, blood stage parasite biomass can be substantial and can reach up to $10^{12}$ infected red blood cells. Asexual blood stage infection, however, is preceded by a small mosquito-transmitted inoculum of ~10–400 Pf sporozoites (SPZ), which specifically invade hepatocytes and replicate therein (Beier et al, 1991). Because this 1-week liver phase of infection (i) is clinically silent and (ii) represents a life-cycle bottleneck, it provides an early target for a malaria vaccine that would prevent blood stage infection with malaria parasites and thereby prevent all disease and transmission stages (Vaughan and Kappe, 2017).

Studies in murine malaria models have been paramount for informing human vaccine trials with live, metabolically active SPZ. Successful and emerging vaccine strategies informed by immunisation and challenge experiments in laboratory mice include (i) irradiated SPZ (Hoffman et al, 2010; Hoffman et al, 2002; Nussenzweig et al, 1967; Seder et al, 2013), (ii) sporozoite infections under chloroquine (CQ) treatment (Beaudoin et al, 1977; Belnoue et al, 2004; Mordmüller et al, 2017; Roestenberg et al, 2009), (iii) sporozoite infections under pyrimethamine treatment (Friesen et al, 2011; Mwakingwe-Omari et al, 2021), and (iv) genetically arrested parasites (Goswami et al, 2024; Lamers et al, 2024; Mueller et al, 2005b; Roozen et al, 2025). Despite the limited ability of murine immunisation models to predict the complex immune responses of humans, the human trials largely reproduced VE documented in the mouse models. Accordingly, an integrative approach combining the entire path of translational research from discovery to pre-clinical experiments to a clinical trial for explorative vaccine strategies is warranted.

Superior VE against Pf malaria as compared to results reported for subunit vaccines has been demonstrated by intravenous inoculation of radiation-attenuated, aseptic, purified, cryopreserved PfSPZ, Sanaria® PfSPZ Vaccine (Epstein et al, 2017; Ishizuka et al, 2016; Lyke et al, 2017; Seder et al, 2013; Sissoko et al, 2017), and intravenous inoculation of infectious, aseptic, purified, cryopreserved PfSPZ (Sanaria® PfSPZ Challenge) with concomitant chemoprophylaxis with CQ, Sanaria® PfSPZ-chemoprophylaxis vaccination (CVac) (CQ) (Bastiaens et al, 2016; Mordmüller et al, 2017; Mwakingwe-Omari et al, 2021). Importantly, initial results indicate higher, per-sporozoite VE of PfSPZ-based vaccine protocols that rely on late chemo-attenuation such as those using the chemoprophylactic drug CQ, which is active only against blood stage parasites (Mordmüller et al, 2017; Roestenberg et al, 2009). Based on this, it has been postulated that the parasite expansion in the liver substantially boosts vaccine potency (Belnoue et al, 2004; Mordmüller et al, 2017; Roestenberg et al, 2009), compared to radiation-attenuated parasites that undergo developmental arrest and death soon after hepatocyte invasion. This strategy, however, requires that adequate drug concentrations are maintained in the blood beyond the liver stage period to kill the parasites when they emerge from the liver, e.g., by weekly administration of the antimalarial drug CQ. It also entails the safety risk of exposing the vaccinee to transient parasitaemia and transient systemic reactions related to malaria 7–9 days after immunisation, mainly fever. To increase the safety margin of infectious PfSPZ immunisations it

would be preferable if the parasites never emerged from the liver without compromising VE. Moreover, chemoprophylaxis that acts against liver stages may substantially strengthen a vaccination regimen with concomitant administration of PfSPZ and chemoprophylaxis and thus, would be a significant step towards a simplified, pragmatic PfSPZ immunisation protocol.

Atovaquone–proguanil (AP) is a licensed drug combination that kills Pf parasites both in red blood cells and in the liver, i.e., before they can emerge into the bloodstream (Davies et al, 1989; Deye et al, 2012; Radloff et al, 1996). Atovaquone disrupts the electron transfer chain in the mitochondrion of the parasite, and resistance to atovaquone arises from a single point mutation in the mitochondrially encoded cytochrome *b* subunit (Blasco et al, 2017). More recently, atovaquone was shown to be also active against Pf mosquito stages (Paton et al, 2019). The exact mode of action of proguanil is less well defined but it acts synergistically with atovaquone (Canfield et al, 1995; Fidock et al, 1998). Currently, when used for chemoprophylaxis, AP (250 mg of atovaquone and 100 mg of proguanil) is administered daily beginning several days before any possible exposure and for 7 days after the last possible exposure. The AP regimen used for the treatment of uncomplicated Pf malaria consists of a 3-day regimen of once daily oral doses of 1000 mg of atovaquone and 400 mg of proguanil. It had previously been shown that a single chemoprophylactic dose of AP (250 mg/100 mg) 1 day before Pf sporozoite challenge via bites of 5 Pf-infected *Anopheles stephensii* mosquitoes prevented blood stage infections (Deye et al, 2012). We thus hypothesised that a single therapeutic dose of AP (1000 mg/400 mg) would be sufficient to fully attenuate even substantially higher doses of PfSPZ during immunisation.

Here, we present the results of pre-clinical experiments and a clinical trial exploring the VE and mechanism of protection of PfSPZ co-administered with a single therapeutic dose of the chemoprophylactic drug combination AP.

## Results

### Exposure to AP leads to early *Plasmodium berghei* (Pb) liver stage arrest

To initiate our pre-clinical analysis, we determined the in vitro effect of a single dose of atovaquone (A) alone or AP when added to cultured hepatoma cells together with Pb SPZ (Fig. 1A). Quantification of liver stage volume and number revealed small, developmentally arrested liver stages (Figs. 1A and EV1). Notably, A- and AP-treated parasites persisted for several days, indicative of developmental arrest rather than immediate cell death (Fig. EV1A). A persistence phenotype is also seen with irradiated SPZ (Scheller and Azad, 1995) and under exposure to pyrimethamine (Friesen et al, 2011), but not with primaquine, which rapidly kills early liver stages (Putrianti et al, 2009). Hepatocyte invasion appeared to be unaffected by drug exposure since the number of intracellular liver stages was indistinguishable from untreated controls (Fig. EV1C). Causal prophylactic activity was confirmed in a murine malaria model (Fig. 1B–D). Administration of a single dose of 3 mg/kg atovaquone plus 1.2 mg/kg proguanil or 3 mg/kg atovaquone alone in C57BL/6 mice inoculated with $10^4$ Pb SPZ prevented blood stage infection in 37 out of 38 mice (97%) (Fig. 1B). Quantitative reverse

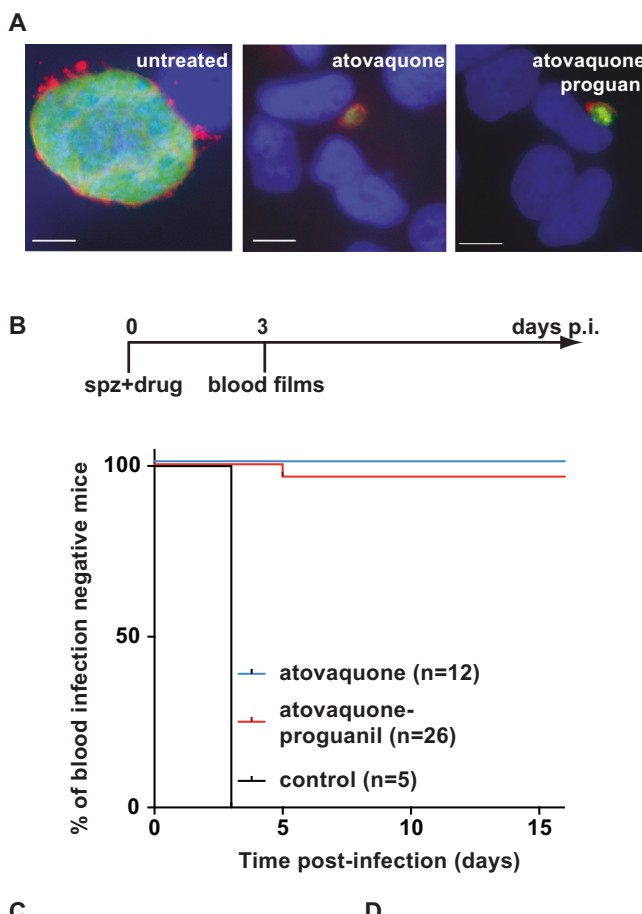

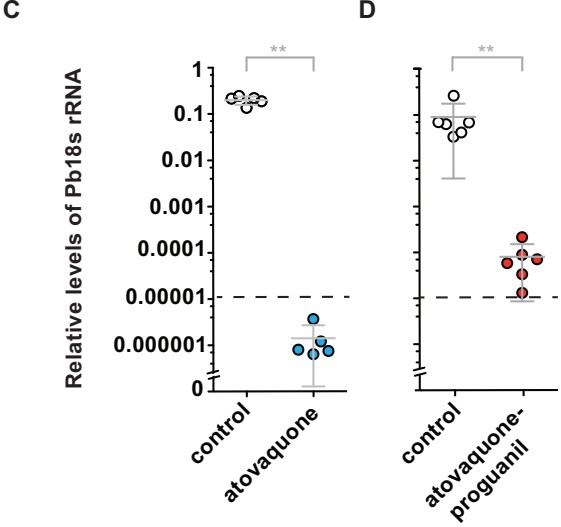

**Figure 1. Early arrest of *Plasmodium berghei* liver stage development after co-administration of live sporozoites and atovaquone or atovaquone–proguanil.**

(A) Composite fluorescence micrographs of *Plasmodium berghei* (Pb) liver stages in cultured hepatoma cells. Shown are representative images of liver stages 48 h after infection with sporozoites. During the first 3 h cultures were exposed to atovaquone (A), atovaquone–proguanil (AP) or buffer with the equivalent amounts of DMSO (0.01%) as a negative control. Parasites were visualised by fluorescent staining of the cytoplasm (green; anti-PbHSP70 antibody), the parasitophorous vacuolar membrane (red; anti-PbUIS4 anti-serum), and nuclei (blue; Hoechst 33342). Scale bars: 10 μm. Data are from a single experiment. (B) Infection and treatment protocol. Shown are the timeline and a Kaplan–Meier analysis of the proportion of C57BL/6 mice that remained blood film-negative after a single intravenous dose of $10^4$ *P. berghei* sporozoites without drug or co-administration of 3 mg/kg A or 3/1.2 mg/kg AP. Control vs. atovaquone, ***$p < 0.0001$; control vs. atovaquone–proguanil, ***$p < 0.0001$; log rank (Mantel–Cox) test. Shown are cumulative data from two (A co-administration, $n = 6$ each) and five (AP co-administration) independent experiments ($n = 3$–6 mice each). Colour code: black line, control ($n = 5$); blue line, atovaquone ($n = 12$); red line, atovaquone–proguanil ($n = 26$). (C, D) Liver parasite load 42 h after infection of C57BL/6 mice with $10^4$ sporozoites and co-administration of (C) 3 mg/kg A or (D) 3/1.2 mg/kg AP or no drug. Shown are mean values (±S.D.) of relative RNA levels of Pb18S rRNA normalised to mouse *GAPDH* ($n \geq 5$). (C) control vs. atovaquone, **$p = 0.0079$; (D) control vs. atovaquone–proguanil, **$p = 0.0022$; Mann–Whitney *U*-test. Colour code: white circles, control ((C) $n = 5$, (D) $n = 6$); blue circles, atovaquone ($n = 5$), red circles, atovaquone–proguanil ($n = 6$). Number and nature of the replicates as well as exact *p* values are shown in Appendix Tables S4 and S5. Source data are available online for this figure.

transcriptase polymerase chain reaction (qRT-PCR) analysis of steady-state levels of Pb18S rRNA and, for comparison, mouse *GAPDH* mRNA in infected livers 42 h after Pb sporozoite and drug co-administration (Fig. 1C,D) revealed a >1000-fold reduced liver stage load in drug-treated mice, in good agreement with our in vitro data (Figs. 1A and EV1). Furthermore, when Pb SPZ were pre-exposed to AP on ice for 2 h, all C57BL/6 mice that received $10^4$ treated ($n = 5$) or untreated Pb SPZ ($n = 3$) developed blood stage infection after 3 days (Fig. EV2A), confirming earlier results of no

direct effect on SPZ (Fowler et al, 1995). Thus, exposure to atovaquone or AP exerts prophylactic activity against Pb liver stage parasites, but not Pb SPZ, allowing maximum hepatocyte invasion followed by robust early liver stage arrest.

## Pb sporozoite immunisation under AP induces sterile protection and robust IFNγ-secreting CD8+ CD11a+ T cell responses

As proof of concept, we next immunised three groups of C57BL/6 mice with i.v. $10^4$ Pb SPZ and concomitant administration of A or AP at weekly intervals (Fig. 2). Mice were challenged 3–4 weeks after the last immunisation dose by intravenous (i.v.) inoculation of $10^4$ Pb SPZ. Three immunisations at weekly intervals with $10^4$ Pb SPZ and atovaquone resulted in sterile protection in all (6/6) mice (Fig. 2A,E). Remarkably, only two immunisations with Pb SPZ and either A or AP still elicited sterile protection in 88% (15/17) of mice and a substantial delay to blood infection in the two mice that were not protected (Fig. 2A). We independently confirmed our findings by quantification of liver stage parasites by qRT-PCR (Fig. 2B). After sporozoite challenge parasite liver loads were reduced by at least two orders of magnitude in immunised animals as compared to controls ($p < 0.05$) (Fig. 2B). Potential residual effects of drug treatment that could have interfered with assessment of VE were ruled out in an independent experiment. In this experiment, AP did not inhibit infections in naive mice when challenged 2 weeks after drug administration (Fig. EV2B).

We next quantified signatures of effector-memory CD8+ T cell responses that correlate with protection against challenge with wild-type SPZ (Berenzon et al, 2003; Cockburn et al, 2011) by measuring IFNγ secretion of CD8+ CD11a+ T cells after

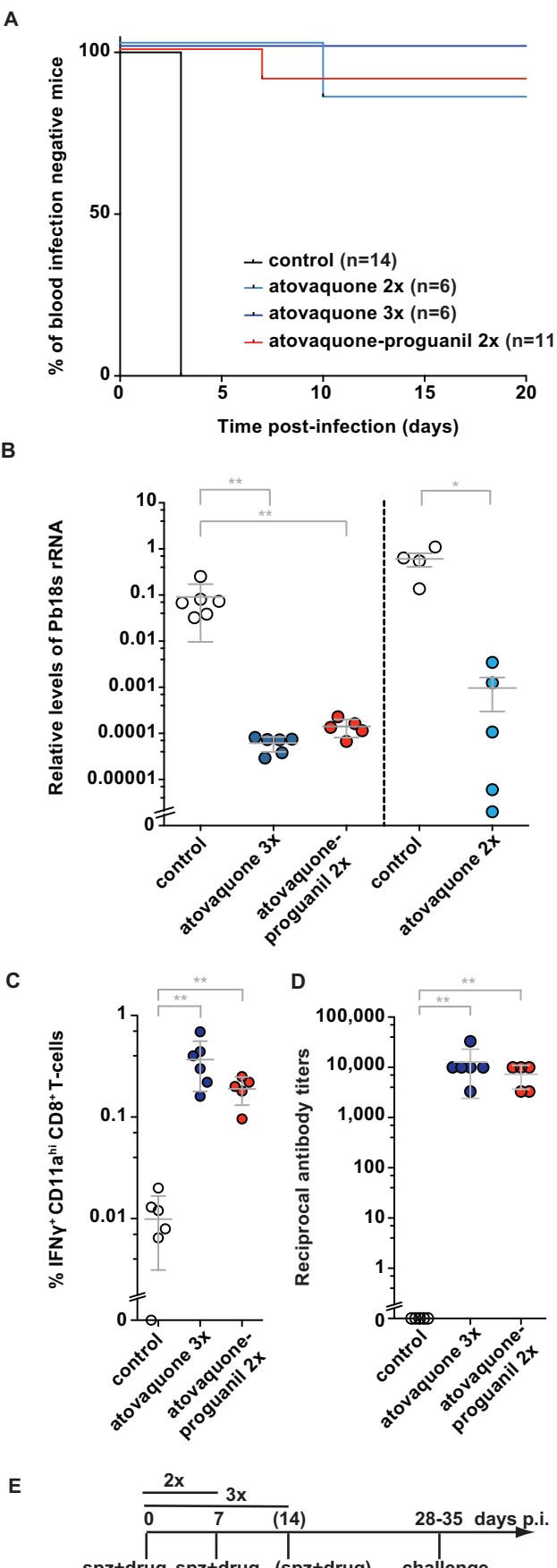

◄ **Figure 2.** **Robust protection against sporozoite challenge infections and antigen-specific immune responses after sporozoite/atovaquone (–proguanil) immunisation.**

(A) Kaplan–Meier analysis of protection in mice immunised by co-administration of sporozoites and a single dose of atovaquone (A; 3 mg/kg i.p.) or atovaquone–proguanil (AP; 3/1.2 mg/kg i.p.). Mice were either immunised twice (AP co-administration; A co-administration) or three times (A co-administration). Naive mice served as controls. Sporozoite challenge was done by i.v. injection of $10^4$ sporozoites 3–4 weeks after the last immunisation. Control vs. atovaquone 3×, ***$p < 0.0001$; control vs. atovaquone 2×, ***$p < 0.0001$; control vs. atovaquone–proguanil, ***$p < 0.0001$; log rank (Mantel–Cox) test. Colour code: black line, control ($n = 14$); blue line, atovaquone 2× ($n = 6$); dark blue line, atovaquone 3× ($n = 6$); red line, atovaquone–proguanil ($n = 11$). (B) Quantification of parasite liver loads after challenge infection. Mice were immunised twice (A co-administration; AP co-administration) and three times (A co-administration) as in (A). Challenge infection was done by i.v. injection of $10^4$ sporozoites at least 3 weeks after the last immunisation. Livers were harvested 42 h later and parasite RNA quantified by qRT-PCR. Relative RNA levels of Pb18S rRNA were normalised to mouse *GAPDH*. Shown are mean values (±S.D.). Control vs. atovaquone 3×, **$p = 0.0022$; control vs. atovaquone–proguanil, **$p = 0.0043$; control vs. atovaquone 2×, *$p = 0.0159$; Mann–Whitney *U*-test. Colour code: white circles, control ($n = 6$ and 4); dark blue circles, atovaquone 3× ($n = 6$), red circles, atovaquone–proguanil ($n = 5$); blue circles, atovaquone 2× ($n = 5$). (C) Quantification of SSP2/TRAP$_{130-138}$ peptide-specific IFNγ secretion by CD8$^+$ CD11a$^+$ T cells from spleens of immunised or control mice. Shown are mean values (±S.D.). Control vs. atovaquone, **$p = 0.0022$; control vs. atovaquone–proguanil, **$p = 0.0043$; Mann–Whitney *U*-test. Colour code: white circles, control ($n = 6$); blue circles, atovaquone ($n = 6$); red circles, atovaquone–proguanil ($n = 5$). (D) Quantification of anti-sporozoite antibody titres from serum of immunised or control mice. Shown are mean values (±S.D.). Control vs. atovaquone, **$p = 0.0043$; control vs. atovaquone–proguanil, **$p = 0.0079$; Mann–Whitney *U*-test. Colour code: white circles, control ($n = 5$); dark blue circles, atovaquone ($n = 6$); red circles, atovaquone–proguanil ($n = 5$). (E) Timeline of the vaccination schedule. Shown are the immunisation intervals and the time point of sporozoite challenge. Data are from a single experiment. Number and nature of the replicates as well as exact *p* values are shown in Appendix Tables S4 and S5. Source data are available online for this figure.

stimulation with the peptides SSP2/TRAP$_{130-138}$ and S20$_{318-326}$, which are recognised in immunised H2-K$^b$-restricted C57BL/6 mice (Hafalla et al, 2013; Müller et al, 2017) (Figs. 2C, EV3, and EV4). Mice were immunised with two doses of Pb SPZ/AP or three doses of Pb SPZ/A (Fig. 2E). To overcome the limitation of low numbers of tissue-resident cytotoxic CD8$^+$ T cells in the liver (Fernandez-Ruiz et al, 2016) we isolated the more abundant splenic antigen-specific CD8$^+$ effector memory (T$_{em}$) cells, which are established for systematic quantitative comparisons of vaccine regimens (Friesen and Matuschewski, 2011; Friesen et al, 2010; Hafalla et al, 2013), 3–4 weeks after the last immunisation, respectively. We detected high levels of antigen-specific IFNγ secretion after both immunisation protocols (Figs. 2C and EV4B–E). Total numbers of effector memory CD8$^+$ CD62L$^{lo}$ T cells were also enhanced after immunisation (Fig. EV4A).

Exposure to sporozoite inoculations activates antibody-producing B cells (Nardin et al, 1998; Offeddu et al, 2012; Rodrigues et al, 1993). Accordingly, the immunisation protocols also induced high (~1:10$^4$) anti-Pb sporozoite antibody titres (Fig. 2D).

In conclusion, the preclinical data demonstrate high chemoprophylactic efficacy of AP leading to reliable, early arrest of liver stage development when co-administered with Pb SPZ. Co-administered AP with intravenous Pb SPZ as part of an immunisation protocol resulted in parasite-specific effector-memory CD8$^+$ T cell responses and robust protection against challenge infections. Of note, VE was markedly superior to another early-arrest protocol with primaquine and indeed, comparable to the most potent chemo-attenuation protocols tested so far (Friesen and Matuschewski, 2011), prompting the design of a clinical trial.

## Clinical trial of PfSPZ-CVac (AP)

Based on the positive pre-clinical results, we conducted a randomised, double-blind, placebo-controlled clinical trial of PfSPZ-CVac (AP) from September 2016 to November 2017 at the Institute for Tropical Medicine in Tübingen, Germany (Malaria controlled human infection trial E, MALACHITE; ClinicalTrials.gov Identifier: NCT02858817). The study population was selected to represent healthy, malaria-naive volunteers aged 18–45 years from Tübingen and the surrounding area. In total, 30 volunteers (15 per dosage/group) were enrolled; 15 in Group A (PfSPZ

reference dose) and 15 in Group B (PfSPZ threefold higher dose). They were randomly allocated to receive three injections of Sanaria® PfSPZ Challenge (aseptic, purified, cryopreserved PfSPZ of the NF54 strain, $n = 10$ in each group) or normal saline placebo ($n = 5$ in each group) per dosage group (Mordmüller et al, 2015; Roestenberg et al, 2013).

In Group A, participants received three doses of $5.12 \times 10^4$ PfSPZ by direct venous inoculation (DVI) at 4-week intervals and oral administration of a single dose of AP (1000 mg/400 mg; 4 times the currently recommended daily chemoprophylaxis dose but equivalent to the daily dose of a 3-day therapeutic regimen in adults) within one hour before each PfSPZ inoculation. In Group B, participants received $1.5 \times 10^5$ PfSPZ with the same schedule and chemoprophylactic regimen. Ten weeks after the last immunisation, participants in both groups underwent controlled human malaria infection (CHMI) by DVI of $3.2 \times 10^3$ PfSPZ of PfSPZ Challenge (NF54).

Twenty-seven participants were included in the per-protocol analysis. Three withdrawals occurred, all of them in Group A before CHMI; one requested by the participant and two withdrawals by the investigators based on non-compliance or non-availability for critical study procedures. An additional participant in Group A was lost to follow-up after CHMI and was included in the per-protocol analysis. This volunteer developed parasitaemia and started treatment on day 12 post-CHMI. Following successful completion of antimalarial treatment, thick blood smear (TBS) and qPCR were negative on day 21 post-CHMI. On day 22 post-CHMI, the volunteer was not reachable and refused further follow-up visits. The study flow chart and baseline population characteristics are detailed in Appendix Fig. S1 and Appendix Table S1.

Importantly, during immunisation no breakthrough parasitaemia by qPCR occurred, demonstrating robust causal prophylactic activity of a single dose of 1000 mg/400 mg AP, even with an inoculum of $1.5 \times 10^5$ PfSPZ. This sporozoite dose exceeds the infective dose of $3.2 \times 10^3$ PfSPZ (Gomez-Perez et al, 2015; Mordmüller et al, 2015) by ~50 times and is estimated to be equivalent to the bites of ~200 infected mosquitoes. Of note, safety and tolerability during immunisation were similar between placebo controls and vaccinees (see Appendix Table S2).

Upon challenge by CHMI, 6 out of 8 vaccinees in group A and all 4 placebo recipients developed Pf parasitaemia (VE, 25%; 95% CI, −12% to 50%). Due to the low efficacy and according to

protocol, no heterologous repeat CHMI was performed. Subsequently, Group B underwent homologous CHMI with PfSPZ Challenge (NF54), i.e., with the vaccine strain only. In Group B, 8 of 10 vaccinees and all 5 placebo controls developed Pf parasitaemia (VE, 20%; 95% CI, −9% to 41%). Moreover, time to qPCR detectable parasitaemia, a marker for partial efficacy, was similar between the groups (Group A; median 7 days; interquartile range (IQR) 7–8.5 and Group B; median 7 days; IQR 7–7.5; Kruskal–Wallis (KW) $H$ test, $p = 0.27$) (Fig. 3).

## Safety of immunisation with PfSPZ-CVac with AP chemoprophylaxis

The overall frequencies of grade 1–4 adverse events (AEs) were similar in recipients of PfSPZ-CVac and normal saline placebo (487 in 20 verum volunteers vs. 221 in 10 placebo volunteers). Of the overall 708 AEs, 483 were considered unrelated or unlikely to be related, 91 as possibly related, 124 as probably related and 8 as definitely related to the administration of PfSPZ Challenge (see Appendix Table S2). The most frequent AE was headache ($n = 81$), followed by an increase in diastolic blood pressure ($n = 61$) and systolic blood pressure ($n = 79$). One serious AE (SAE) occurred after the first vaccination: a female patient was hospitalised 17 days after the first vaccination with left lower abdominal pain and diagnosed with an ovarian cyst rupture. This SAE was deemed to be unrelated to the study. In total 531 grade 1, 132 grade 2, 40, grade 3 and 5 grade 4 AEs were recorded. All grade 4 AEs (including the above-described SAE with ovarian cyst rupture and associated abdominal pain, hypoglycaemia and two episodes of creatinine kinase elevation) were considered as unrelated or unlikely to be related to the study treatment. The mild local and systemic symptoms after immunisation (tenderness, pruritus of the injection site, headache, fatigue, dizziness, laboratory changes) were transient and resolved within a few days.

Of the 27 individuals who underwent CHMI, 23 developed Pf parasitaemia. Twenty one of these 23 experienced at least one mild or moderate (grade 1 or 2) symptom associated with Pf parasitaemia or antimalarial treatment. Four related grade 3 AEs occurred: lymphocytopenia, thrombocytopenia, fever, subcostal pain. All study-associated AEs resolved completely without sequelae.

## Anti-PfCSP antibodies generated by early arrest of PfSPZ-CVac immunisation

The 25% and 20% VE observed after 3 doses of $5.12 \times 10^4$ or $1.5 \times 10^5$ PfSPZ of PfSPZ-CVac (AP), respectively, contrasts sharply with the 100% VE we achieved with 3 doses of $5.12 \times 10^4$ PfSPZ of PfSPZ-CVac (CQ) (Mordmüller et al, 2017). To better understand the immunological basis for these differences in VE, we first analysed the levels of IgG antibodies generated by vaccination against the Pf circumsporozoite protein (PfCSP) 2 weeks after the third dose of PfSPZ-CVac (AP) vaccine and just prior to CHMI (Fig. 4A). The median concentration of PfCSP-specific IgG antibodies as estimated by enzyme-linked immunosorbent assay (ELISA), 1 day before CHMI was 4 arbitrary units (AU) [range: 1.7–81 AU] for the 8 subjects undergoing CHMI in vaccine Group A and 35 AU [range 4.6–396 AU] for the 9 subjects who underwent CHMI in the PfSPZ-CVac (CQ) study group (Mordmüller et al, 2017), which used the same dose of

$5.12 \times 10^4$ PfSPZ ($p = 0.0026$, analysis of variance (ANOVA) post hoc test) (Mordmüller et al, 2017). Since we can firmly exclude batch-to-batch variation during the manufacturing process by GMP-mandated quality control procedures, which include quantification of sporozoite numbers, motility and invasion capacity, this finding is consistent with a scenario in which the immune systems of subjects immunised with PfSPZ-CVac (AP) are exposed to less PfCSP than those immunised with PfSPZ-CVac (CQ).

Next, we assessed anti-PfCSP IgG antibodies in the higher PfSPZ dose group of PfSPZ-CVac (AP). Increasing the three PfSPZ doses to $1.5 \times 10^5$ PfSPZ of PfSPZ-CVac (AP) triggered significantly higher median anti-PfCSP antibody concentrations 1 day before CHMI compared to the $5.12 \times 10^4$ PfSPZ-CVac (AP) (38 AU, range: 5.3–170 AU, $p < 0.0001$, ANOVA post hoc test) (Fig. 4A). However, despite the >9-fold increase in PfCSP antibodies in group B there was no increase in VE. For PfCSP-specific IgM antibodies determined by ELISA, a similar pattern was seen. Vaccination using PfSPZ (AP) led to approx. fivefold lower antibody levels as PfSPZ (CQ) using the same dose ($5.12 \times 10^4$ PfSPZ) ($p < 0.0001$), but when the dose of PfSPZ (AP) was increased to $1.5 \times 10^5$ PfSPZ, the PfCSP-specific IgM response reached similar levels as achieved with $5.12 \times 10^4$ PfSPZ PfSPZ-CVac (CQ) (Fig. 4B).

We note that the VE of 20% achieved with $5.12 \times 10^4$ PfSPZ of PfSPZ-CVac (AP) was significantly lower than the 100% VE after immunisation with the same $5.12 \times 10^4$ PfSPZ dose of PfSPZ-CVac (CQ) ($p < 0.001$, Barnard's test, two-tailed). Strikingly, the higher levels of anti-PfCSP antibody concentrations in Group B were not predictive of VE after CHMI (Fig. 4A,B). These results are thus consistent with the anti-PfCSP antibody-independent, cellular immunity-based, mechanism of protection after immunisation with PfSPZ-CVac (CQ) found in mice for CVac (CQ) and non-human primates for radiation attenuated SPZ (Doolan and Hoffman, 2000; Schofield et al, 1987; Weiss and Jiang, 2012; Weiss et al, 1988).

## Profiling of IgG antibody responses to sporozoite, liver stage and asexual blood stage antigens

We hypothesised that reduced protection with PfSPZ-CVac (AP) compared to PfSPZ-CVac (CQ) was due to the early arrest of liver stage development by AP. This pharmaceutical arrest is expected to result in significantly reduced exposure to liver stage and blood stage antigens. To this end, we probed a representative range of IgG antibody responses with a custom protein microarray (Doolan et al, 2008). We detected a striking absence of three distinct antibody responses in volunteers who received PfSPZ-CVac (AP) compared to PfSPZ-CVac (CQ) (Mordmüller et al, 2017), while the breadth and intensity of immunoreactivity to 216 other antigens was indistinguishable (Fig. 5A,B). In agreement with the ELISA results (Fig. 4A), IgG responses to PfCSP were considerably higher in volunteers with high-dose PfSPZ-CVac (AP) compared to low-dose PfSPZ-CVac (AP) (Fig. 5C), likely reflecting higher exposure to SPZ but not protection. Most importantly, IgG antibody responses to the two well-known secreted liver stage antigens LISP2 and LSA1 were significantly reduced in recipients of PfSPZ-CVac (AP) (Fig. 5B,C, Appendix Table S3). Using the protein microarrays, we detected higher IgG concentrations against PfMSP5 in PfSPZ-CVac (CQ)-vaccinated individuals compared to PfSPZ-CVac (AP)-vaccinated individuals (Fig. 5B,C). This prompted us to establish an

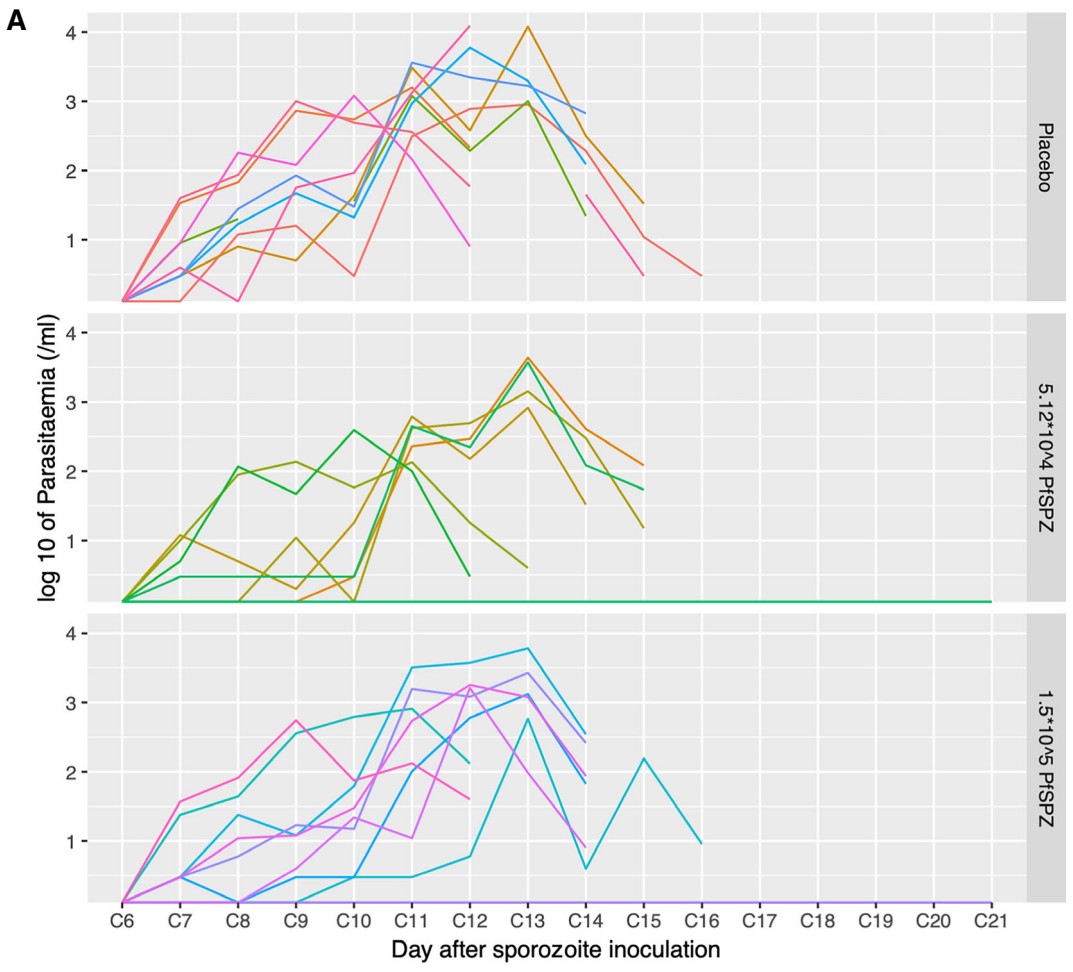

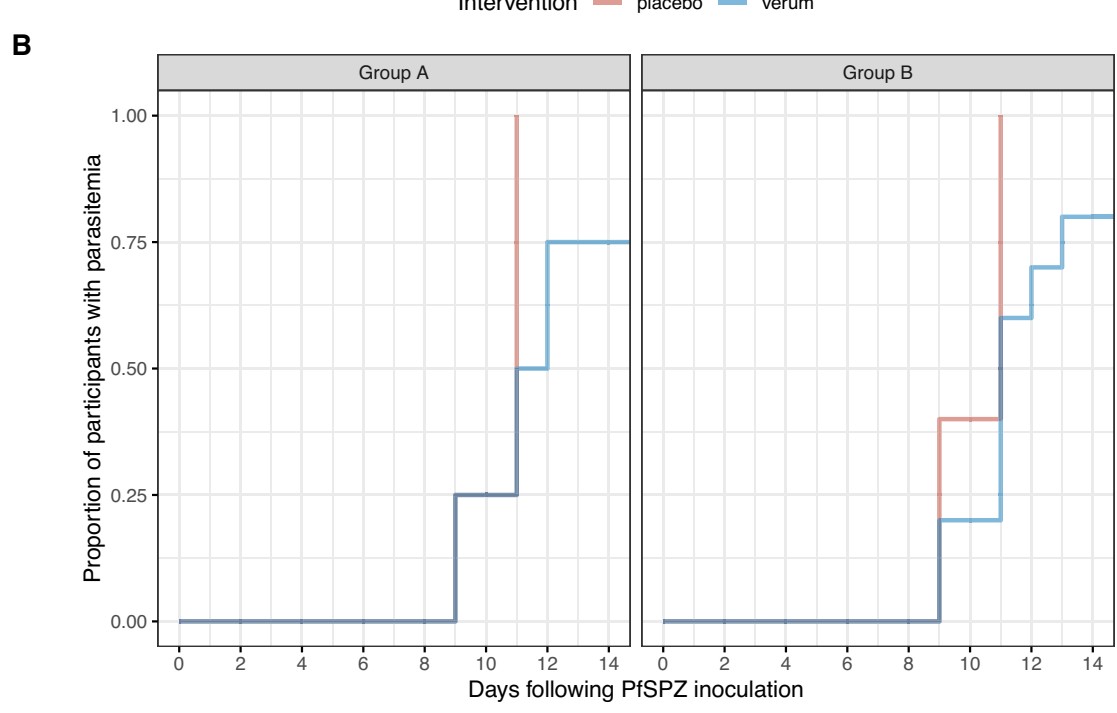

**Figure 3. qPCR-assessed Pf parasitaemia kinetics and time to patency in clinical trial participants after CHMI.**

(A) Shown is the quantification of Pf blood stage parasite load ($\log_{10}$ per ml blood) by qPCR over time after CHMI in the nine individuals from the placebo control group (top), the six non-protected individuals in group A (reference PfSPZ vaccine dose; centre) and the eight non-protected individuals in group B (high PfSPZ vaccine dose; bottom). Curves in different colours depict parasite densities over time in individual participants. Treatment was initiated upon reaching the pre-defined parasitaemia endpoint. (B) Shown are Kaplan–Meier curves of time to initiation of treatment upon reaching a pre-defined parasitaemia endpoint. Groups A and B received three doses of $5.12 \times 10^4$ or $1.5 \times 10^5$ PfSPZ/AP, respectively. CHMI was done at week 10 after the last immunisation in all volunteers, with the exception of one volunteer in Group A (verum) and B (placebo) each, who underwent CHMI at 17 and 14 weeks, respectively. Both were treated for blood infections on days 10 and 12 after CHMI and were included in the graph. Source data are available online for this figure.

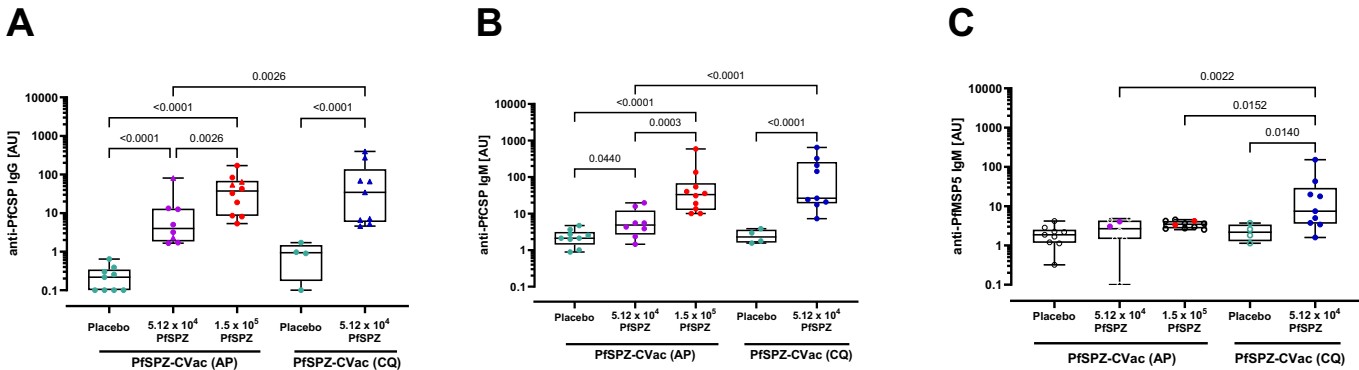

**Figure 4. Antibodies to PfCSP and PfMSP5 1 day before CHMI within the current trial (PfSPZ-CVac [AP]) compared to data from a previous trial (PfSPZ-CVac [CQ]) (Mordmüller et al, 2017).**

Antibody levels were estimated by ELISA. For the antigen CSP, both IgG (A) as well as IgM (B) antibody levels are presented. For the MSP5 antigen (C), IgM antibody levels are shown. Data are presented by boxplots (median with 25th and 75th quartile levels, full range from minimum to maximum). Filled triangles represent individuals who did not develop Pf parasitaemia (protected), and filled circles represent individuals who did develop Pf parasitaemia (unprotected). Note that anti-PfCSP antibody levels did not predict VE during CHMI. For quantitative analysis, reactivity was estimated in comparison to a coated human IgG or IgM standard, respectively. Colour code: turquoise: placebo ($n = 9$ for PfSPZ-CVac [AP]; $n = 4$ for PfSPZ-CVac [CQ]); violet, $5.12 \times 10^4$ PfSPZ-CVac [AP] ($n = 8$); red, $1.5 \times 10^5$ PfSPZ-CVac [AP] ($n = 10$); and blue, $5.12 \times 10^4$ PfSPZ [CQ] ($n = 9$). All ELISA experiments were performed in three technical replicates. Source data are available online for this figure.

ELISA assay using recombinant purified PfMSP5. We detected significant IgG antibody reactivity against PfMSP5 only in one donor within the PfSPZ-CVac (CQ) study group (Fig. EV5). However, we were able to detect significantly increased IgM anti-PfMSP5 antibody responses in the PfSPZ-CVac (CQ)-vaccinated individuals, with a notable absence of such responses in the PfSPZ-CVac (AP)-vaccinated study population (Fig. 4C).

Together, our IgG profiling data showed that inferior VE of the PfSPZ-CVac (AP) protocol correlated with low or absent responses to several liver stage antigens. In contrast, levels of ELISA-determined IgG responses to PfCSP in PfSPZ-CVac (AP) immunised individuals were not predictive of protection as they had been in a previous study with PfSPZ-CVac (CQ) (Sulyok et al, 2021).

## Discussion

Intra-host replication is a key feature of many live attenuated vaccines. This expansion correlates with improved protective immunity against natural exposure (Minor, 2015). For instance, the Sabin poliomyelitis vaccine proved that a polio virus strain capable of replicating in the gut but not the nervous system can generate robust neutralising anti-viral immunity (Minor, 2015). Similarly, it has been proposed that intra-host replication of live, attenuated, PfSPZ-based vaccines might substantially boost the per-

parasite VE (Bastiaens et al, 2016; Borrmann and Matuschewski, 2011a, b; Butler et al, 2011; Friesen et al, 2010; Mordmüller et al, 2017; Richie et al, 2015). Clinical trials of two early arresting PfSPZ vaccines, PfSPZ Vaccine (radiation-attenuation) and PfSPZ-CVac (PYR) (chemo-attenuation) have shown protection against CHMI with heterologous parasites (Epstein et al, 2017; Lyke et al, 2017; Mordmuller et al, 2022; Mwakingwe-Omari et al, 2021). In contrast, a trial of an early arresting PfSPZ vaccine, PfSPZ-GA1 (attenuation by deletion of two genes) (Roestenberg et al, 2020) did not give significant protection against homologous CHMI. The large differences in VE are unexpected, since arrests occur at very similar time points in liver stage development, prior to onset of intracellular replication. It also highlights that malaria vaccine development has been largely empirical, and a better understanding of the critical processes involved in the establishment of protective immunity could be useful to guide further rational development of PfSPZ vaccines.

In this combined preclinical and clinical study, we tested the potential of an attenuation protocol based on AP, a licensed antimalarial drug combination with liver stage activity used for the treatment and chemoprophylaxis of Pf malaria. We proposed that AP would improve the in vivo chemo-attenuation strategy significantly because: (i) chemoprophylaxis would be administered concomitantly with PfSPZ increasing safety and practicality, whereas CQ prophylaxis starts prior to PfSPZ injection and is continued with weekly doses until after the last vaccination; (ii)

**A**

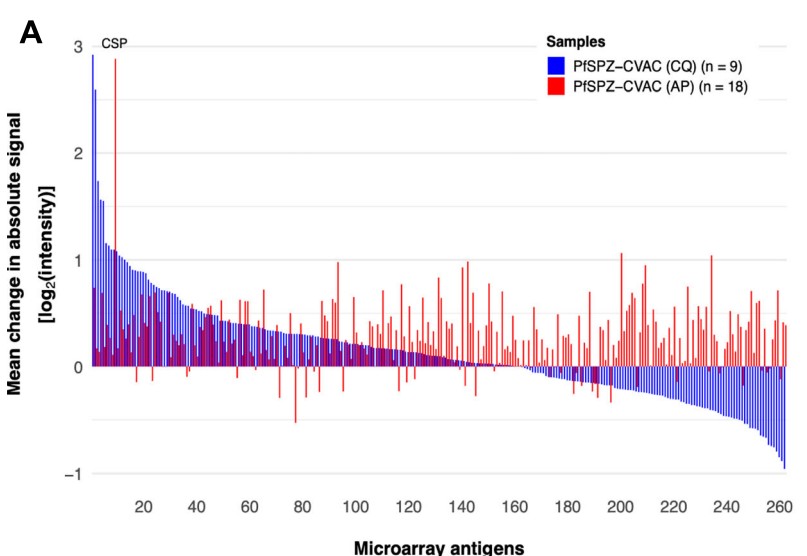

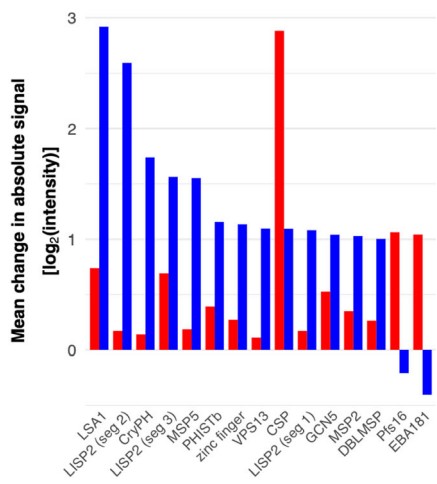

**B**

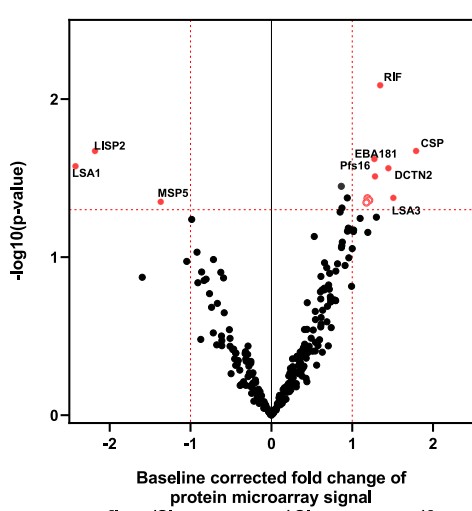

**C**

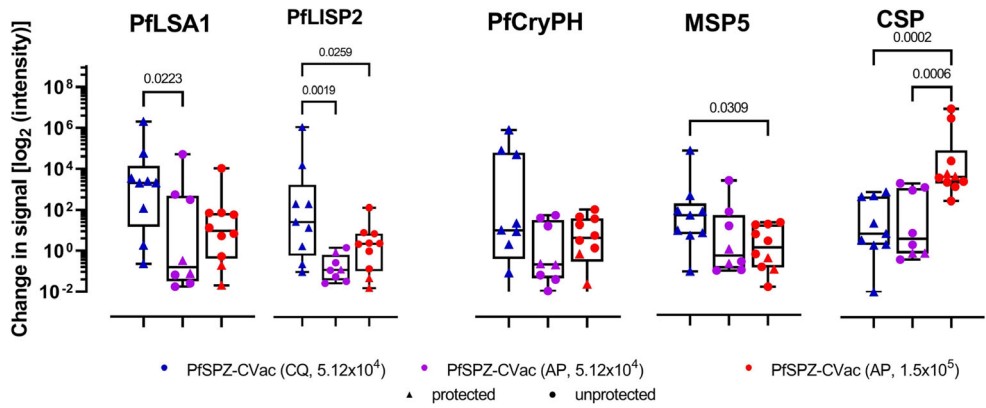

◄ **Figure 5. Antibody response measured by protein microarrays comparing reactivities of the two vaccination regimen PfSPZ-CVac (AP) and PfSPZ-CVac (CQ).**

Sera from all volunteers collected before immunisation (baseline, D-1) and 1 day before challenge (C-1) were applied on protein microarrays at a 1:50 dilution containing 262 Pf proteins representing 228 unique antigens. Analysis was performed on C-1 data after subtraction of the individual baseline reactivity. (A) Bar charts of mean reactivity in the PfSPZ-CVac (AP, doses $5.12 \times 10^4$ PfSPZ and $1.5 \times 10^5$ grouped together) and the PfSPZ-CVac (CQ, $5.12 \times 10^4$ PfSPZ) regimen, ordered by descending signal intensities for PfSPZ-CVac (CQ) of IgG against all antigens (left panel) and subset of highest mean changes (>2-fold change, right panel) from both vaccine protocols. Array data were normalised, $\log_2$-transformed and baseline reactivity was subtracted. (B) Volcano plot to analyse differential immunoreactivity in the two trials. Antigen reactivity in verum donors of PfSPZ-CVac (AP, doses $5.12 \times 10^4$ PfSPZ and $1.5 \times 10^5$ grouped together) was compared to the verum donors of PfSPZ-CVac (CQ) (dose $5.12 \times 10^4$ PfSPZ). The fraction of the two reactivities are shown. Differentially recognised antigens (Student's $t$ test; $p$ value < 0.05 and fold change >2) are depicted in red. (C) Box plot (median with 25th and 75th quartile levels, full range) of signature IgG responses in PfSPZ-CVac (CQ) (dose $5.12 \times 10^4$ PfSPZ) and PfSPZ-CVac (AP) (stratified by vaccine dose) and. Each data point represents the results of an individual study participant. Colour code: blue, $5.12 \times 10^4$ PfSPZ [CQ] ($n = 9$); violet, $5.12 \times 10^4$ PfSPZ-CVac [AP] ($n = 8$); and red, $1.5 \times 10^5$ PfSPZ-CVac [AP] ($n = 10$). Filled circles: unprotected individuals; filled triangles: protected individuals. Numbers indicate the results of the Freund's LSD post hoc test (Graph-Pad Prism v10.4.1). The alpha-level was set to 0.05. Source data are available online for this figure.

there would be no egress of the parasites from the liver, and thus, no possibility of malaria symptoms associated with transient asexual erythrocytic stage parasitaemia, which occurs regularly with doses beyond $10^5$ PfSPZ of PfSPZ-CVac (CQ); and (iii) AP is in general, better tolerated and safer than CQ.

We demonstrated in the murine malaria model that atovaquone alone and AP led to an early arrest of liver stage development. Compared to untreated controls, drug-exposed intrahepatic Pb parasites did not begin to replicate as evidenced by single-nucleated liver stage forms observed in vitro and by >100-fold reduced parasite liver loads. Of significant clinical importance, this robust liver arrest was confirmed in the subsequent clinical trial, in which a single dose of AP completely prevented blood stage infections after each of three immunisations with a high dose of $1.5 \times 10^5$ PfSPZ of PfSPZ Challenge. By comparison, the dose of PfSPZ Challenge that leads to consistent infection is only $3.2 \times 10^3$ PfSPZ (Mordmüller et al, 2015). In contrast to chemo-attenuation with CQ, we did not detect any asexual blood stage infection, even by ultra-sensitive qPCR, during immunisation with doses of PfSPZ Challenge of $1.5 \times 10^5$ PfSPZ (total number of PfSPZ challenge = 58; 100% infectious dose) under AP.

Previously, full protection was elicited by three doses of $5.12 \times 10^4$ PfSPZ and chemo-attenuation with CQ (7), which only kills Pf parasites that have emerged from the liver and have commenced subsequent intraerythrocytic replication. In stark contrast, only 2 out of 8 volunteers, who received three doses of $5.12 \times 10^4$ PfSPZ with AP chemoprophylaxis, were protected against CHMI with $3.2 \times 10^3$ PfSPZ of PfSPZ Challenge. Even a threefold increase in the PfSPZ numbers per dose to three doses of $1.5 \times 10^5$ PfSPZ failed to induce significant protection, unlike previous results from PfSPZ-based vaccines, which displayed high-level VE upon dose increases, including in radiation, pyrimethamine-arrested, and CQ-attenuated PfSPZ immunisation protocols (Mordmüller et al, 2017; Mwakingwe-Omari et al, 2021; Seder et al, 2013). An attractive hypothesis is that persistence of early arrested parasites, as documented in murine malaria models for irradiated SPZ (Scheller and Azad, 1995) and likely also occurring during pyrimethamine arrest (Friesen et al, 2011; Jiang et al, 1988) translates into better local priming of memory cytolytic T cell responses. It will be interesting to compare the dynamics of CD8+ T cell priming, including by monocyte-derived CD11c+ cells (Kurup et al, 2019) in these vaccination strategies.

Due to the inaccessibility of liver-resident immune cells and as a proxy for protective immune responses (Plotkin, 2008), we quantified the serum antibody profiles in protected and susceptible

human volunteers with a custom protein microarray. Levels of anti-PfCSP antibodies on the day of CHMI in volunteers immunised with three doses of $1.5 \times 10^5$ PfSPZ were at least as high as levels induced by three immunisations with $5.12 \times 10^4$ PfSPZ in the PfSPZ-CVac (CQ) group previously reported (Mordmüller et al, 2017). However, we observed a considerable reduction of IgG antibody responses to the two well-characterised liver stage antigens LISP2 and LSA1, neither of which antibodies were correlated with protection in a previous trial of PfSPZ-CVac (CQ) (Sulyok et al, 2021). Using ELISA, we detected higher levels of IgM, but not IgG, antibodies recognising MSP5 after vaccination with PfSPZ-CVac (CQ) as compared to PfSPZ-CVac (AP). This is reminiscent of previous observations with early arresting gamma-irradiated PfSPZ vaccination (PfSPZ Vaccine), which also elicited a highly focused IgM antibody response to MSP5 in protected vaccinees (Tumbo et al, 2024). Importantly, unlike our earlier study (Mordmüller et al, 2017), the comparison of similar PfSPZ doses of PfSPZ-CVac (AP) and PfSPZ-CVac (CQ) allowed us to entangle the effects of PfSPZ dose and liver stage maturation, resulting in a clear separation of responses that appear to reflect sporozoite exposure (anti-PfCSP IgG) from responses associated with biological processes that may be critical for protection (anti-PfLISP2 and anti-PfLSA1 IgG). Taken together, our data indicate that high-level VE (≥80%) 10 weeks after the last immunising dose induced by immunisation with 3 doses of $5.12 \times 10^4$ to $1.5 \times 10^5$ infectious PfSPZ under chemoprophylaxis appears to depend on intra-hepatic replication of PfSPZ.

We cannot formally exclude that AP has a heretofore undefined immunosuppressive activity in humans as reported for CQ (Chen et al, 2018). We consider this notion unlikely since, in more than 20 years of clinical use of AP, no such evidence has been reported. The observed discrepancy between the VE achieved with sporozoite immunisation under AP in the murine and human studies (full vs. non-significant minimal protection, respectively) is possibly related to the shorter duration of the Pb liver stage (~2 days) compared to the Pf liver stage (~7 days). A longer duration is expected to amplify any differential impact of the time point of arrest on VE. We also noted a largely reduced anti-PfCSP antibody response measured by ELISA, but not with the protein array, in group A vaccinees in comparison to an identical ($5.12 \times 10^4$) PfSPZ dose in PfSPZ-CVac (CQ) vaccinees (Mordmüller et al, 2017). A plausible explanation is that PfCSP is produced and acts as an immunogen not only in extracellular PfSPZ but also throughout Pf liver stage development, coherent with robust CSP expression during liver stage maturation in murine malaria models (Mueller et al, 2005a;

Singh et al, 2007). Accordingly, the full liver stage maturation achieved by PfSPZ-CVac (CQ) immunisation as compared to lack of maturation after PfSPZ-CVac (AP) immunisation is a likely cause for the marked differences in anti-PfCSP antibody levels measured by ELISA after immunisation with the same total number of PfSPZ.

In our clinical trial of PfSPZ-CVac (AP) protection appeared to be unrelated to the high anti-PfCSP antibody levels in this study, contrary to our previous study with PfSPZ-CVac (CQ) (Sulyok et al, 2021). It has been previously argued that the correlation between ELISA-measured anti-PfCSP antibody levels and protection to CHMI (Sulyok et al, 2021) and natural exposure (Sissoko et al, 2017; Sissoko et al, 2022) is indicative of vaccine, i.e., sporozoite dose, but not of a protective immunological mechanism. This is in line with murine and non-human primate studies showing that the protection induced by radiation-, chemo-, and genetically attenuated SPZ immunisation is strictly dependent on CD8+ T cells (Epstein et al, 2011; Friesen et al, 2010; Schmidt et al, 2008). Although direct activity of human CD8+ T cells against Pf liver stages in humans has not been demonstrated, our data support the hypothesis that cellular immune mechanisms are central to VE of chemo-attenuated PfSPZ vaccines. Further immunological studies are warranted to elucidate the mechanistic basis and identify immune correlates of vaccine-induced protection in humans.

Our results support an essential role for intrahepatic replication by late attenuation for achieving maximum protection against CHMI. This conclusion is strengthened by results from two very recent phase 1/2a trials with the GA2 vaccine (Lamers et al, 2024; Roozen et al, 2025). GA2 lacks the PfMEI2 gene and arrests late in the liver stage, around 6 days after liver cell invasion of SPZ (Hafalla et al, 2025). Notably, nearly complete protection was achieved with only a single round of vaccination with GA2 (Roozen et al, 2025).

In conclusion, we have two major findings regarding PfSPZ vaccines and one finding regarding the chemoprophylactic efficacy of a single dose of AP. First, there was a disconnect between the complete protective efficacy of PbSPZ-CVac (AP) in mice and the lack of protective efficacy of PfSPZ-CVac (AP) in humans. Second, we demonstrated that immunisation with the PfSPZ-CVac (CQ) (blood stage arresting, fully replicating) induced a greater magnitude and breadth of antibodies than did immunisation with the same dose of PfSPZ-CVac (AP) (early arresting, non-replicating). These findings support the contention that the full protection of PfSPZ-CVac (CQ) vs. the minimal protection of PfSPZ-CVac (AP) is due to immune responses against the antigens expressed during liver stage replication. Our analytical approach allowed us to delineate responses that are only bystanders, including the association between anti-sporozoite surface antibodies and protection, and responses that we interpret as bona fide immunological correlates of a critical biological process, in this case, full liver stage maturation. Finally, a single dose of AP that is four times the dose used for daily chemoprophylaxis but equivalent to a daily dose of a 3-day therapeutic regimen gave complete causal prophylactic protection against an excessive Pf sporozoite challenge. We think that further studies to assess this higher dose of AP at a less than daily dose frequency as a more practical approach to chemoprophylaxis with AP are warranted.

# Methods

## Reagents and tools table

| Reagent/resource | Reference or source | Identifier or catalogue number |
| --- | --- | --- |
| **Experimental models** | | |
| Huh-7 | Human hepatoma cells | Nakabayashi et al (1982) |
| C57BL/6J mice | Charles River | RRID:IMSR_JAX:000664 |
| NMRI mice | Charles River | RRID:IMSR_CRL:605 |
| Anopheles stephensi SK | Feldmann and Ponnudurai, 1989; PMID: 2519646 | Feldmann and Ponnudurai (1989) |
| Plasmodium berghei clone 507 | Janse et al, 2006; PMID: 16242190 | Janse et al (2006) |
| PfSPZ | Sanaria | GMP-produced Plasmodium falciparum NF54 sporozoites for phase I clinical trial |
| **Recombinant DNA** | | |
| – | – | – |
| **Antibodies** | | |
| Anti-mouse CD8a-PerCP-Cy5.5 (53-6.7) | Thermo Fisher Scientific | Cat# 45-0081-82, RRID: AB_1107004 |
| Anti-mouse CD11a-e450 (M17/4) | Thermo Fisher Scientific | Cat#48-0111-82, RRID: AB_11064445 |
| Anti-mouse CD62L-PE-Cy7 (MEL-14) | Thermo Fisher Scientific | Cat# 25-0621-82, RRID: AB_469633 |
| Anti-mouse IFN-γ-APC (XMG1.2) | Thermo Fisher Scientific | Cat# 17-7311-82, RRID: AB_469504 |
| Rabbit polyclonal anti-Plasmodium berghei UIS4 | Müller et al, 2011; PMID: 21673790 | Müller et al (2011) |
| Mouse anti-Plasmodium berghei HSP70 | Tsuji et al, 1994; PMID: 8153120 | RRID:AB_2650482 |
| Alexa Fluor 488 goat anti-mouse | Thermo Fisher Scientific | Cat # A-11029 RRID: AB_2534088 |
| Alexa Fluor 546 goat anti-mouse | Thermo Fisher Scientific | Cat # A-11003 AB_2534071 |
| Alexa Fluor 546 goat anti-rabbit | Thermo Fisher Scientific | Cat # A-11010 RRID:AB_2534077 |
| Goat anti-human IgG-HRP | Jackson ImmunoResearch | #109-035-098 |
| Goat anti-human IgM-HRP | ImmunoReagents | #GtxHu-006-E2HRPX |
| Human IgG | ThermoFisher | #31154 |
| Human IgM | Sigma-Aldrich | #I8260 |
| Goat anti-human IgG QDot™ 800 | Grace Bio-Labs | #110635 |
| Biotin-SP-conjugated goat anti-human IgM | Jackson ImmunoResearch | #109-065-043 |
| **Oligonucleotides and other sequence-based reagents** | | |
| P. berghei 18SrRNA (GI, 160641) forward: 5′-AAGCATTAAATAAAGCGAATACATCCTTAC-3′; reverse: 5′-GGAGATTGGTTTTGACGTTTATGTG-3′ | Friesen et al, 2010; PMID: 20630856 | Friesen et al (2010) |
| Mouse GAPDH gene (GI, 281199965) forward: 5′-CGTCCCGTAGACAAAATGGT-3′; reverse: 5′-TTGATGGCAACAATCTCCAC-3′ | Friesen et al, 2010; PMID: 20630856 | Friesen et al (2010) |
| **Chemicals, enzymes and other reagents** | | |
| Acetone | Roth | Cat # 9780.2 |
| Brefeldin A | eBioscience/Invitrogen | Cat # 00-4506-51 |
| Dimethyl sulfoxide (DMSO) | PanReac AppliChem | Cat # A3672,0100 |
| DMEM | Gibco/Thermo Fisher Scientific | Cat # 11965092 |
| Foetal Bovine Serum (South America) | Gibco/Thermo Fisher Scientific | Cat # 10500064 |
| Fluoromount-G | Southern Biotech, Biozol | Cat # 0100-01 |

| Reagent/resource | Reference or source | Identifier or catalogue number |
|---|---|---|
| Giemsa | Sigma-Aldrich/ Merck | Cat # 1.09204.0500 |
| Hoechst 33342 | Invitrogen/ Thermo Fisher Scientific | Cat # H3570 |
| Lab-Tek 8 | Nunc/Thermo Fisher Scientific | Cat # 177410PK |
| MicroAmp 96-well RT-PCR plates | Applied Biosystems/ Thermo Fisher Scientific | Cat # N8010560 |
| Paraformaldehyd | Sigma-Aldrich | Cat # 44124-1KG |
| PBS | Gibco/Thermo Fisher Scientific | Cat # 10010023 |
| Penicillin-Streptomycin | Gibco/Thermo Fisher Scientific | Cat # 15140122 |
| Perm-Wash Buffer | BD | Cat # 554723 |
| Power SYBR Green: PCR Master-Mix | Applied Biosystems/ Thermo Fisher Scientific | Cat # 4367659 |
| Retroscript Kit | Ambion/Thermo Fisher Scientific | Cat # AM1710 |
| RNeasy Mini Kit | QIAGEN | Cat # 74104 |
| RNAse-Free DNAse Set | QIAGEN | Cat # 79254 |
| RPMI 1640 | Gibco/Thermo Fisher Scientific | Cat # 11875093 |
| TRIzol reagent | Invitrogen/ Thermo Fisher Scientific | Cat # 15596026 |
| Qdot™585 streptavidin conjugate | Invitrogen | #Q10111MP |
| **Software** | | |
| FlowJo 8.7 | FlowJo, 385 Williamson Way Ashland, OR 97520 USA | https://www.flowjo.com/ |
| Fiji 2.1.0/1.53c | Image J, NIH, Bethesda, USA | https://fiji.sc |
| GraphPad Prism 9 | GraphPad Prism 225 Franklin Street Boston, MA 02110 USA | https:// www.graphpad.com |
| **Other** | | |
| Flow cytometer (LSRII) | BD | N/A |
| DM2500 light/fluorescence microscope | Leica | N/A |
| Axio Vision microscope | Zeiss | N/A |
| Quantitative PCR instrument: StepOnePlus ABI 7500 sequence detection system | Applied Biosystems | N/A |

## Clinical trial

This trial was approved by the ethics committee of the Eberhard Karls University and the University Hospital Tübingen as well as by the Paul Ehrlich Institute (Langen, Germany) and the Regional Council (Regierungspräsidium Tübingen). The study was compliant with the International Council for Harmonisation Good Clinical Practice guidelines, the German Medicinal Product Act (Arzneimittelgesetz, AMG), the World Medical Association (WMA) Declaration of Helsinki and the Department of Health and Human Services Belmont Report. The trial was registered at ClinicalTrials.gov (NCT02858817). Informed consent was obtained from all study participants prior to screening.

This single-centre, double-blind, randomised, placebo-controlled phase 1 clinical trial was conducted from September 2016 to November 2017 at the Institute of Tropical Medicine in Tübingen, Germany. The study population was selected to represent healthy, malaria-naive adults. Volunteers aged 18–45 years from Tübingen and the surrounding area with a body mass index (BMI) between 18 kg/m$^2$ and 30 kg/m$^2$ were included. Female participants were required to practice effective contraception and to provide a negative pregnancy test. Further inclusion criteria included: being reachable at all times by mobile phone during the whole study, agreement to share medical information about the volunteer with his or her general practitioner and understanding of study procedures and risks, assessed by a quiz. Additionally, willingness to undergo CHMI with PfSPZ Challenge, to take a curative regimen of antimalarial if necessary, and the ability to comply with all study requirements (in the investigator's opinion) were also required.

Exclusion criteria were: a history of malaria or plans to travel to endemic regions during the study, receiving any investigational product in another clinical trial within 90 days before enrolment or planned receipt during the study, previous participation in a malaria vaccine trial, history of serious psychiatric conditions, convulsions, or severe head trauma, any malignancy, and diabetes mellitus. Moreover, falling in moderate risk or higher categories for fatal or non-fatal cardiovascular event within 5 years (van Meer et al, 2014), prolonged QTc interval (>450 ms), or any other clinically significant abnormalities in the electrocardiogram, breast feeding, or intention to become pregnant, HIV, hepatitis B or C virus infection, alcohol or drug abuse, any suspected immunodeficient state, history of splenectomy, and haemoglobinopathies also prevented participation. A complete list of eligibility criteria is available as an online supplement. Eligibility criteria were assessed after written informed consent was given.

Fifteen volunteers per group were enrolled and randomised to receive Sanaria® PfSPZ Challenge for immunisation or normal saline placebo with an allocation ratio of 2:1 for vaccine:placebo. Syringes containing either PfSPZ or placebo were prepared by a separate formulation team and both clinical investigators and study participants were blinded to intervention allocation. In Group A, participants received $5.12 \times 10^4$ PfSPZ of PfSPZ Challenge (NF54) by DVI three times at 4-week intervals and a single dose of AP (1000 mg/400 mg) administered orally within 1 h before each immunisation. In Group B, participants received $1.5 \times 10^5$ PfSPZ of PfSPZ Challenge (NF54) by DVI with the same scheduling and chemoprophylactic regimen.

Ten weeks after the last immunisation, the first CHMI was performed in both groups for VE testing. CHMI was done by DVI of $3.2 \times 10^3$ PfSPZ of PfSPZ Challenge (NF54). Active follow-up of the participants was conducted for 56 days after the injection of PfSPZ Challenge for CHMI. The protocol stipulated two successive CHMIs, the first at 10 weeks and the second at 16–44 weeks. The sequence of CHMI was planned to be PfSPZ Challenge (NF54, homologous clone) followed by PfSPZ Challenge (7G8, heterologous clone) for Group A. The sequence for Group B was to be based on VE following first CHMI in Group A: NF54 followed by 7G8 when VE against homologous CHMI was <75%, 7G8 followed by 7G8 when VE was ≥75%. Due to the low efficacy of the first CHMI also in group B a second CHMI was not performed in

Group B. First CHMI was thus performed with PfSPZ Challenge (NF54) for both groups.

PfSPZ Challenge (NF54) is comprised of aseptic, purified, cryopreserved NF54 PfSPZ, produced by Sanaria Inc. (Rockville, US). PfSPZ Challenge was stored and transported in liquid nitrogen vapour phase at −150 to −196 °C. Formulation and reconstitution were made in Tübingen on the day of infection. Volunteers were inoculated within 30 min after thawing of PfSPZ Challenge. Sterile isotonic normal saline, identical in appearance to PfSPZ Challenge was used as a placebo. A volume of 0.5 ml of vaccine or placebo was injected into an arm vein by DVI through a 25 gauge needle. After each immunisation, participants were monitored for at least 60 min before leaving the clinic for local and systemic AEs. Participants were assessed on site for safety and to measure parasitaemia on days 1, 5, 7, 10, 14 and 21 after each immunisation, and on day 1, 6–21, 28, 56 after CHMI. Medically qualified study personnel were available continually for unscheduled visits. Antimalarial treatment, according to the German guidelines (Deutsche Gesellschaft für Tropenmedizin und Internationale Gesundheit (DTG), 2016) for the treatment of uncomplicated Pf malaria (AP or artemether–lumefantrine as first-line drugs) was to be initiated, in the event of breakthrough parasitaemia with symptoms during immunisation or in the case of parasitaemia following CHMI. Breakthrough parasitaemia was defined as microscopically detectable parasitaemia during immunisation with at least two symptoms consistent with malaria for 2 days despite chemoprophylaxis. Protection was defined as the absence of parasites in the peripheral blood for 28 days following CHMI. Parasitemia after CHMI was assessed on a daily basis from day 6 to day 21 and again on day 28 via TBS and qPCR. Treatment was administered upon occurrence of three consecutive positive PCR results one of them at least 100 Pf parasites/ml from samples taken at least 12 h apart or the first TBS positivity. An additional follow-up visit for safety was conducted on day 56. Participants were encouraged to immediately report AEs between the scheduled follow-up visits.

If a volunteer was withdrawn from the study after receiving a dose of PfSPZ Challenge at one or more of the three immunisations or at CHMI, a full, appropriate, curative course of antimalarial therapy was administered.

In both groups, 10 volunteers were randomly allocated to receive immunisations with PfSPZ Challenge and 5 volunteers to receive normal saline placebo (PfSPZ Challenge (NF54):placebo = 2:1). Group membership was allocated using a Mersenne-Twister random number generator implemented in R. A third party outside the study team generated and distributed the randomisation list. A dedicated member of the formulation team, who was not involved in volunteer management or diagnostic activities, kept the randomisation envelopes and dosing schedule.

The primary aim of the study was to assess the safety and VE of repeated immunisation by DVI of PfSPZ Challenge under AP chemoprophylaxis in malaria naive adults. The primary VE endpoint was the proportion of protected volunteers.

Protection was defined as the absence of parasites in the peripheral blood for 28 days following first CHMI with PfSPZ Challenge. To assess safety outcomes, Grade 3 and 4 AEs and SAEs were captured from time of first administration of A/P until the end of the study. Functional characterisation of humoral and cellular immune responses was an exploratory endpoint.

## Statistical analysis and power calculation of a clinical trial

To be able to show, with a power of 80% and a two-tailed alpha of 5%, that 25% or less of immunised volunteers and 95% of controls, allocated in a 2:1 ratio became infected by CHMI, 10 immunised and 5 placebo-treated volunteers per group were required. Hence, a total of 30 (10 each for $5.12 \times 10^4$ and $1.5 \times 10^5$ PfSPZ Challenge (NF54) with AP and 10 placebo) volunteers were required. The sample size was calculated using the nBinomial function in the gsDesign package.

No formal hypothesis testing was done for safety and tolerability data. Safety and tolerability data are presented as descriptive analyses in listings and graphically. VE was calculated by comparison of proportions between immunised and placebo-treated volunteers using an unconditional exact test (Boschloo's test) and time-to-parasitaemia using a log-rank (LR) test. Multiple non-parametric group comparisons were performed by using the KW *H* test. The level of significance was set at a two-tailed type 1 error alpha <5%. All statistical analyses were performed using R version 3.4.4 and GraphPad Prism v. 10.41. Kaplan–Meier curves were compared by a log-rank (Mantel–Cox) test. Statistical significance of parasite sizes, qPCR data and flow cytometric analyses were assessed using the Mann–Whitney *U*-test for nonparametric test samples. *p* values of $p < 0.05$ were considered significant.

## Animal experiments

All animal work was conducted in accordance with the German Animal Welfare Act (Tierschutzgesetz in der Fassung vom 18. Mai 2006, BGBl. I S. 1207), which implements the directive 86/609/EEC from the European Union and the European Convention for the protection of vertebrate animals used for experimental and other scientific purposes. The protocol was approved by the ethics committee of MPI-IB and the Berlin state authorities (LAGeSo Reg# G0469/09, G0294/15). Female C57BL/6 and NMRI mice at the age of 6–8 weeks were purchased from Charles River (Sulzfeld, Germany) for sporozoite injections and blood stage passages, respectively.

## Pb parasites

For all experiments, Pb ANKA clone 507, which constitutively expresses the green fluorescent protein (GFP) under control of the *eIF1α* promoter (Janse et al, 2006), was used. To generate SPZ, female *Anopheles stephensi* mosquitoes were infected by a blood-meal on Pb-infected NMRI mice. Starting 17 days after infection, salivary glands were hand-dissected from infected mosquitoes, gently ground, and SPZ harvested after centrifugation. Freshly dissected Pb SPZ ($10^4$) were added to complete DMEM medium (10%FCS, 1% Pen/Strep) containing atovaquone (0.2 μM; Wellvone® Suspension, 750 mg/5 ml; GlaxoSmithKline) or Malarone® (0.2 μM atovaquone, 0.08 μM proguanil hydrochloride; GlaxoSmithKline) or the equivalent amounts of DMSO (0.01%) as a negative control. The stock solutions were prepared in 1% DMSO/PBS to increase solubility. Irradiated Pb SPZ and untreated Pb SPZ served as controls. SPZ were added to cultured Huh7 cells in

duplicates. Hepatoma cells were incubated for 1 h at RT for sporozoite sedimentation and for an additional 2 h at 37 °C to permit sporozoite entry. Infected cultures were subsequently washed repeatedly with DMEM complete medium to remove residual drug. Next, infected cells were incubated in complete DMEM medium for 48, 96 or 116 h at 37 °C before fixation in 4% para-formaldehyde. Fixed cells were stained with a monoclonal anti-PbHSP70 antibody (Tsuji et al, 1994) to visualise parasites, with a rabbit anti-PbUIS4 serum (Montagna et al, 2014) to visualise the parasitophorous vacuole, and with Hoechst 33342 (Invitrogen) that stains nuclei. As secondary antibodies, goat Alexa Fluor 488–labelled antibody to mouse immunoglobulin G (IgG) and goat Alexa Fluor 546–labelled antibody to rabbit immunoglobulin G (IgG) (Invitrogen) were used. Image analysis and quantification were performed by fluorescence microscopy using either a Leica DM2500 or a Zeiss Axio Vision microscope. Images were processed with Fiji (Image J, NIH, Bethesda, USA).

## Pb sporozoite infection, immunisation, and challenge experiments

To test whether single-dose administrations of atovaquone or AP prevent a subsequent Pb blood stage infection and thus, life-threatening pathology, C57BL/6 mice were intravenously (i.v.) infected with $10^4$ Pb SPZ, and one dose of drug was co-administered intraperitoneally (i.p.). Drug doses were 3 mg/kg atovaquone alone or 3 mg/kg atovaquone plus 1.2 mg/kg proguanil hydrochloride. Three days later, infections were monitored daily by microscopic examination of Giemsa-stained blood films. To test the toxicity of AP on SPZ, SPZ were treated with 1 μM atovaquone–0.4 μM proguanil–hydrochloride in complete DMEM for 2 h and then washed with RPMI followed by inoculation of $10^4$ treated Pb SPZ to naive mice. For quantification of pre-erythrocytic parasite development by qRT-PCR, livers were removed 42 h after sporozoite infection and transferred to TRIZOL® for RNA isolation. Complementary DNA (cDNA) synthesis and qRT-PCR were done as described previously (Friesen et al, 2010). Briefly, the mean $C_t$ value of the Pb18S ribosomal subunit RNA (18SrRNA; gene ID: 160641) was normalised to the mean $C_t$ of mouse GAPDH mRNA values (gene ID: 281199965) using the $\Delta\Delta C_t$ method. qPCR experiments were performed with the ABI 7500 sequence detection system and were done in triplicate. For immunisations, female C57BL/6 mice were inoculated twice or three times at 7-day intervals with $10^4$ Pb salivary gland SPZ i.v. together with one dose of the drug administered i.p. as described for the chemoprophylaxis studies. Three to 4 weeks after the last immunisation, mice were challenged by intravenous injection of $10^4$ Pb salivary gland SPZ. Naive, age-matched mice served as infection controls. For the determination of sterile protection, blood parasitaemia was monitored by microscopic examination of Giemsa-stained blood films daily from day 3 until day 20. Alternatively, the parasite liver load after challenge was quantified by qRT-PCR.

## Anti-sporozoite antibody titres

For titration of Pb-specific antibodies in mouse serum, salivary gland-associated SPZ were deposited on glass slides, air-dried and fixed in acetone. After rehydration in PBS and blocking, mouse serum was titrated by serial dilutions and bound antibodies

detected with a secondary goat anti-mouse Alexa Fluor 546-coupled antibody (1:1000). Nuclei were stained with Hoechst 33342 (1:1000).

## Enzyme-linked immunosorbent assay

IgG antibodies to the PfCSP and Pf merozoite surface protein 5 (PfMSP5) were assessed by ELISA as previously described (Pinilla et al, 2021; Sulyok et al, 2021; Tumbo et al, 2024). 96-well plates (Costar 96-well microtitre high binding plates) were coated overnight at 4 °C either with 0.5 μg/ml of the recombinant PfCSP protein or recombinant PfMSP5 protein in 100 μl coating buffer. Plates were washed three times with 1× PBS, 0.1% Tween 20 (PBST) and blocked with blocking solution (Candor, #11050) for PfCSP and with phosphate-buffered saline with tween (PBST) supplemented with 5% bovine serum albumin (BSA) and 0.5 mM EDTA for PfMSP5. Plates were washed three times and in blocking buffer serially diluted serum samples (in duplicate) were added and incubated at room temperature (RT) for 1 h. After three washes, peroxidase labelled goat anti-human IgG (Jackson Immuno Research, #109-035-098) was added at a dilution of 1:10,000 and incubated at RT for 1 h. Plates were washed four times, TMB peroxidase substrate was added for plate development, and the plates were incubated for 15 min at 22 °C RT. Reactions were stopped with 1 M HCl stop solution. For the estimation of antigen-specific IgM antibodies, the same procedure was applied, with the exception that the blocking buffer was 1× RotiBlock (Roth) and the secondary antibody was HRP-conjugated goat anti-human IgM (ImmunoReagents, #GtxHu-006-E2HRPX) at a dilution of 1:5000. The plates were immediately read with a CLARIOstar microplate reader (BMG) at 450 nm. Data analysis was with Clariostar Software Version 5.40 R2 and MARS Data Analysis Software Version 3.31. A negative control (pooled serum from non-immune individuals from a malaria-free area) was included in all assays. Serum of a pool of individuals with anti-PfCSP or PfMSP5 antibodies was used as positive control. Antibody concentrations in AU were estimated by a respective dilution series of highly pure human IgG (ThermoFisher, #31154) or highly pure IgM (Sigma, #I8260), which was precoated on separate wells on the same plates. To estimate the relative concentration and standardise the difference between the plates for IgG and IgM, a four-parameter logistic curve was fitted to the results of IgG and IgM positive control standards and the respective concentrations in the sera using the R statistical software package version 4.4.3. Data were analysed and figures generated using GraphPad Prism v. 10.41. To test for statistical differences, one-way ANOVA was used, and $p$ values were estimated using the two-stage step-up method of Benjamini, Krieger and Yekutieli. The alpha-level was set to 0.05.

## Protein microarray and analysis

Microarrays were produced at the University of California Irvine, Irvine, California, USA (Doolan et al, 2008). In total, 262 Pf proteins and protein domains were expressed using an E. coli lysate in vitro expression system and spotted on a 16-pad ONCYTE AVID slide, representing 228 important Pf antigens known to frequently appear in sterile and naturally acquired immunity against the parasite (Kobayashi et al, 2019; Obiero et al, 2019).

Protein microarray measurements were performed as previously described (Pinilla et al, 2021; Sulyok et al, 2021). For the detection of binding antibodies, secondary IgG antibody (goat anti-human IgG QDot™800, Grace Bio-Labs #110635), secondary IgM antibody (biotin-SP-conjugated goat anti-human IgM, Jackson ImmunoResearch #109-065-043) and Qdot™585 streptavidin conjugate (Invitrogen #Q10111MP) were used. Study serum samples as well as a European control serum pool were diluted 1:50 in 0.05× Super G Blocking Buffer (Grace Bio-Labs, Inc.) containing 10% *E. coli* lysate (GenScript, Piscataway, NJ) and incubated for 30 min on a shaker at RT. Meanwhile, microarray slides were rehydrated using 0.05× Super G Blocking buffer at RT. Rehydration buffer was subsequently removed and samples were added onto the slides. Arrays were incubated overnight at 4 °C on a shaker (180 rpm). Serum samples were removed the following day and microarrays were washed using 1× TBST buffer (Grace Bio-Labs, Inc.). Secondary antibodies were then applied at a dilution of 1:200 and

---

**The paper explained**

**Problem**

A significantly more effective and longer lasting malaria vaccine than current generation of vaccines (RTS,S/AS01 and R21/Matrix-M) is needed to move from control of malaria caused by *Plasmodium falciparum* (Pf) towards elimination and eventually, eradication. The most efficacious malaria vaccine approach so far is immunisation with attenuated, whole malaria sporozoites (SPZ), the parasite stage injected by mosquitoes, and the best protection has come from immunisation with PfSPZ attenuated by co-administration of an anti-malarial drug, PfSPZ-CVac. The time point of attenuation in the liver appears to be critical for both optimal protection and maximum safety. Based on successful experiments in rodents, we tested the antimalarial combination atovaquone/proguanil (AP) as a new attenuation method for PfSPZ-CVac.

**Results**

We showed in a phase I clinical trial in human volunteers that even a single dose of AP was sufficient to fully arrest Pf liver stage infections after inoculations with up to $1.5 \times 10^5$ PfSPZ and thus, to reliably prevent life-threatening malaria blood stage infections with Pf. However, the immunisation protocol consisting of direct venous inoculations of live PfSPZ in combination with AP (PfSPZ-CVac (AP)) elicited only low-level protection against controlled human malaria infection with PfSPZ identical to those in the vaccine. By systematic comparative immune profiling we were able to differentiate antibody responses related to parasite exposure from immunological correlates of parasite maturation in the liver, which boosts vaccine efficacy.

**Impact**

This study showed that PfSPZ-CVac (AP) regimens were not protective against PfSPZ challenge, despite analogous regimens being fully protective in mice. Immunisation with PfSPZ-CVac with chloroquine attenuation (PfSPZ-CVac (CQ) induced stronger antibody responses to liver-stage antigens than PfSPZ-CVac (AP). These findings suggest that protection with PfSPZ-CVac (CQ) is linked to immune responses against liver-stage antigens and that specific antibodies could serve as biomarkers in future clinical trials of PfSPZ-CVac. A single high dose of AP (1000 mg atovaquone/400 mg proguanil) given on the day of PfSPZ inoculation fully blocked blood stage infection, even at high PfSPZ doses. This contrasts with current chemoprophylaxis regimens requiring daily dosing before and after exposure. Our results suggest that fourfold higher, less frequent dosing—potentially weekly—may be effective if started just before exposure.

---

incubated for 2 h at RT on the shaker, followed by another washing step and a one-hour incubation in a 1:250 dilution of Qdot™585 streptavidin conjugate. After a final washing step, slides were dried by centrifugation at $500 \times g$ for 10 min. Slide images were taken using the ArrayCAM® Imaging System (Grace Bio-Labs) and the ArrayCAM 400-S Microarray Imager Software. Microarray data was analysed in R statistical software package version 3.6.2. All images were manually checked for any noise signal. Each antigen spot signal was corrected for local background reactivity by applying a normal-exponential convolution model (Silver et al, 2009) using the "saddle" -algorithm for parameter estimation (available in the limma package v3.28.14) (McGee and Chen, 2006). Data was $log_2$-transformed and further normalised by subtraction of the median signal intensity of mock expression spots on the particular array to correct for background activity of antibodies binding to *E. coli* lysate. Differential antibody levels in the different allocated study outcomes (placebo, non-protected and protected vaccinees) were detected by Student's *t* test. Antigens with $p < 0.05$ and a fold change >2 of mean signal intensities were defined as differentially recognised between the tested sample groups. R software v. 4.4.3 and GraphPad Prism v. 10.41 were used to analyse data and generate figures. To test for statistical differences between the vaccination groups, one-way ANOVA was performed, and *p* values were calculated using Fisher's Least Significance Difference test for multiple comparison between all groups. The alpha-level was set to 0.05.

## Quantification of antigen-specific CD8[+] T cell responses

For CD8[+] T cell stimulations followed by intracellular cytokine staining, splenic lymphocytes were stimulated with 10 μM SSP2/TRAP$_{130-138}$ or S20$_{318-326}$ peptides (Hafalla et al, 2013) for 5 h at 37 °C in the presence of brefeldin A (1:1000). Cells were stained with fluorescently-labelled anti-mouse CD8 [XMG1.2], CD62L [MEL14], or CD11a [M17/4] antibodies (eBioscience). Following fixation with 4% paraformaldehyde, cells were stained intracellularly with fluorescently labelled anti-mouse IFN-γ [R4-6A2] antibody (eBioscience) in permeabilization buffer (BD Bioscience). Antibodies were incubated 60 min at 4 °C. After washing and transfer to 1% paraformaldehyde/PBS, cells were quantified using an LSRII flow cytometer (BD Bioscience). Data analysis was performed using FlowJo 8.7 (Tree Star Inc., Oregon, USA).

## Data availability

This study includes no data deposited in external repositories.

The source data of this paper are collected in the following database record: biostudies:S-SCDT-10_1038-S44321-025-00301-8.

## Peer review information

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

## Acknowledgements

We wish to acknowledge the volunteers participating in the clinical trial. We also gratefully acknowledge the contribution of the manufacturing, quality, regulatory and clinical teams at Sanaria. Funding for the clinical trial was provided by the German Centre for Infection Research (DZIF). We are grateful for the support provided in part by the National Institutes of Allergy and Infectious Diseases, National Institutes of Health (USA), SBIR grants 5R44AI055229 and 2R44AI058375 to SLH. The preclinical work was supported by a DFG grant (SFB 544) to SB and KM and by the Max Planck Society to KM. The clinical trial was funded by AKF grant 346-0-0. SB is supported by the following grants: DAAD 57592740 Global Health Centre, DFG BO 2494/3-2 and DFG BO 2494/7-1, DZIF TTU 03.913 and EDCTP3 101145698.

## Author contributions

**Steffen Borrmann**: Conceptualisation; Data curation; Supervision; Funding acquisition; Writing—original draft; Project administration. **Zita Sulyok**: Investigation; Writing—review and editing. **Katja Müller**: Investigation; Writing—review and editing. **Rolf Fendel**: Supervision; Methodology; Writing—review and editing. **Mihály Sulyok**: Investigation; Writing—review and editing. **Johannes Friesen**: Investigation. **Albert Lalremruata**: Investigation. **Thaisa Lucas Sandri**: Investigation. **The Trong Nguyen**: Investigation. **Carlos Lamsfus Calle**: Investigation. **Annette Knoblich**: Investigation. **Stephanie Sefried**: Investigation; Writing—review and editing. **Javier Ibáñez**: Investigation. **Freia-Raphaella Lorenz**: Formal analysis; Investigation. **Henri Lynn Heimann**: Investigation; attempts to contact contributing author Henri-Lynn Heimann before publication were unsuccessful; the author could not be informed of the article's publication. **David M Weller**: Investigation. **Regina Steuder**: Investigation. **Selorme Adukpo**: Investigation; Project administration. **Patricia Granados Bayon**: Investigation. **Zsófia Molnár**: Investigation; Project administration. **Meral Esen**: Resources; Investigation. **Wolfram Metzger**: Resources; Investigation. **Eric R James**: Resources. **Adam Ruben**: Resources. **Yonas Abebe**: Resources. **Sumana Chakravarty**: Resources. **Anita Manoj**: Resources. **Natasha KC**: Resources; Investigation; Methodology. **Tooba Murshedkar**: Resources; Investigation. **Julius C R Hafalla**: Investigation; Methodology; Project administration; Writing—review and editing. **Tamirat Gebru Woldearegai**: Investigation. **Fiona O'Rourke**: Conceptualisation; Resources; Methodology; Project administration; Writing—review and editing. **Jana Held**: Resources; Validation; Investigation; Methodology. **Pete Billingsley**: Conceptualisation; Resources; Methodology; Writing—review and editing. **B Kim Lee Sim**: Conceptualisation; Resources; Supervision; Funding acquisition; Validation; Methodology; Project administration; Writing—review and editing. **Thomas L Richie**: Conceptualisation; Resources; Supervision; Funding acquisition; Methodology; Project administration; Writing—review and editing. **Stephen L Hoffman**: Conceptualisation; Supervision; Funding acquisition; Methodology; Project administration; Writing—review and editing. **Peter G Kremsner**: Conceptualisation; Data curation; Formal analysis; Supervision; Funding acquisition; Investigation; Visualisation; Methodology; Project administration; Writing—review and editing. **Kai Matuschewski**: Conceptualisation; Supervision; Funding acquisition; Methodology; Project administration; Writing—review and editing. **Benjamin Mordmüller**: Conceptualisation; Data curation; Formal analysis; Supervision; Investigation; Visualisation; Methodology; Project administration; Writing—review and editing.

Source data underlying figure panels in this paper may have individual authorship assigned. Where available, figure panel/source data authorship is listed in the following database record: biostudies:S-SCDT-10_1038-S44321-025-00301-8.

## Funding

## Disclosure and competing interests statement

ERJ, YA, SC, NKC, TM, BKLS, TLR and SLH are all full-time employees of Sanaria Inc., Rockville, MD, the developer and owner of PfSPZ products. PFB,

AR and AM were full-time employees of Sanaria Inc. at the time the work was performed. BKLS and SLH have financial interests in Sanaria Inc. and are named inventors of patents related to Purified *Plasmodium* and Vaccine compositions.

# Expanded View Figures

**Figure EV1.  *Plasmodium* liver stages treated with atovaquone–proguanil retain their invasive capacity, are fully arrested, and persist 5 days after infection in vitro.**

(A) Composite fluorescence micrographs of mature *Plasmodium berghei* liver stages in cultured hepatoma cells. Shown are representative images of liver stages 48, 96, and 116 h after infection with sporozoites. During the first 3 h, cultures were exposed to atovaquone–proguanil. Irradiated, untreated sporozoites served as a control. Parasites were visualised by fluorescent staining of the cytoplasm (green; anti-*Pb*HSP70 antibody), the parasitophorous vacuole membrane (red; anti-*Pb*UIS4 anti-serum), and nuclei (blue; Hoechst 33342). Scale bars: 10 µm. (B) Quantification of liver stage volumes after prophylactic drug treatment. Parasite volume was quantified 48 h after infection and normalised to the average volume of untreated parasites. Shown are mean percentages (±S.D.). Atovaquone, ***$p < 0.0001$; atovaquone–proguanil, ***$p < 0.0001$; Mann–Whitney *U*-test. Colour code: white circles, untreated ($n = 41$); blue circles, atovaquone ($n = 44$); red circles, atovaquone–proguanil ($n = 41$). (C) Sporozoite invasion is unaffected by short-term drug treatment. Shown are mean numbers of liver stages (±S.D.) in cultured hepatoma cells 48 h after initial 3 h co-administration of live sporozoites and atovaquone or atovaquone–proguanil. Untreated sporozoites served as controls. ns, non-significant. Atovaquone, ns $p = 0.6667$; atovaquone–proguanil, ns $p = 0.3333$; Mann–Whitney *U*-test. Colour code: white bar, untreated ($n = 2$), blue bar, atovaquone ($n = 2$); red bar, atovaquone–proguanil ($n = 2$). All in vitro liver stage experiments were performed in two technical replicates. Data in (A–C) are from single biological experiments performed with two technical replicates. Number and nature of the replicates as well as exact *p* values are shown in Appendix Tables S4 and S5. Source data are available online for this figure.

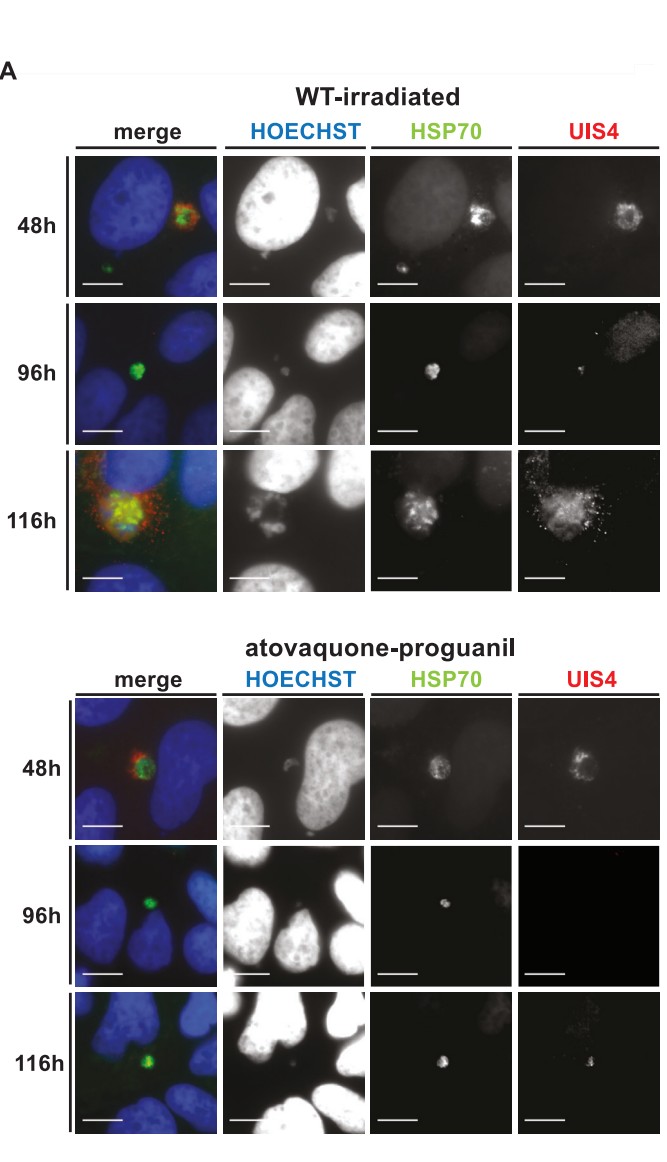

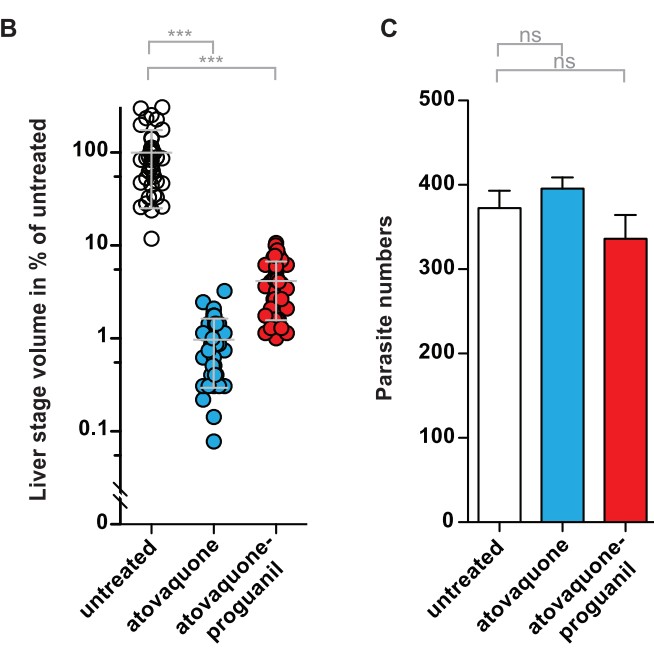

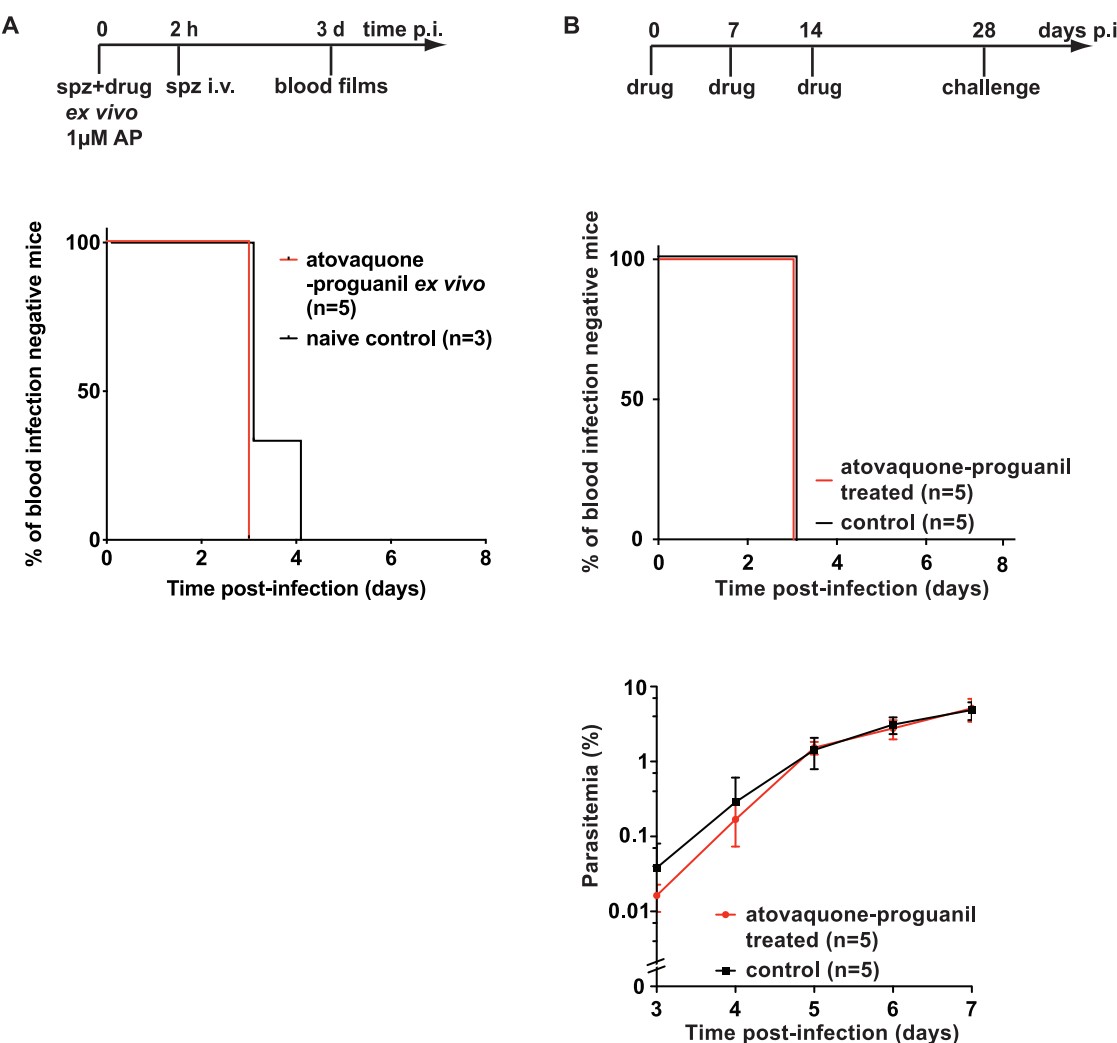

**Figure EV2. Sporozoites exposed to AP retain infectivity and no interference from chemo-attenuation with atovaquone–proguanil with subsequent assessment of protection.**

(A) Pre-exposure of Pb sporozoites to AP on ice for 2 h does not inhibit mouse infection. Shown are the study design (top) and Kaplan–Meier analysis of time to blood infection. ns, non-significant. Control vs. atovaquone ex vivo, ns $p = 0.1967$; log rank (Mantel-Cox) test. Colour code: black line, control ($n = 3$); red line, atovaquone–proguanil ($n = 5$). (B) Atovaquone–proguanil does not inhibit infections in naive mice when challenged 2 weeks after drug administration. C57BL/6 mice were treated three times at weekly intervals with 3/1.2 mg/kg atovaquone–proguanil (top). Kaplan–Meier analysis of time to blood infection upon challenge with $10^4$ sporozoites (centre). ns, non-significant. Control vs. atovaquone-treated, ns $p > 0.99$; log rank (Mantel-Cox) test. Kinetics of blood stage infections after challenge (bottom). Parasitaemia was determined by daily microscopic examination of Giemsa-stained blood films. Shown are mean asexual blood stage parasite densities (±S.D.). Colour code: black line, control ($n = 5$); red line, atovaquone–proguanil ($n = 5$). Number and nature of the replicates as well as exact $p$ values are shown in Appendix Tables S4 and S5. Source data are available online for this figure.

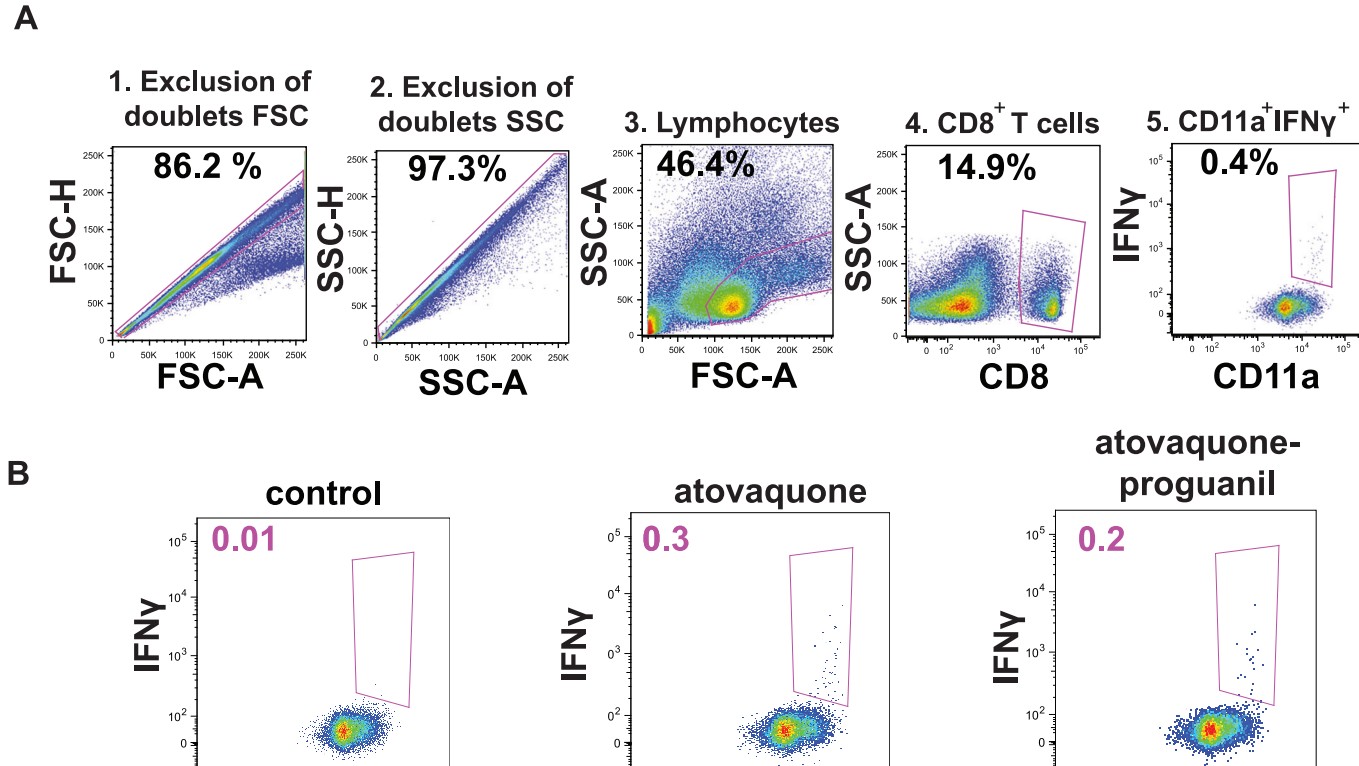

**Figure EV3. Gating strategy for the identification of IFNγ-secreting activated CD8⁺ CD11aʰⁱ T cells after re-stimulation with SSP2/TRAP₁₃₀₋₁₃₈ peptide.**

(A) Shown are representative FACS plots illustrating the gating strategy for IFNγ secretion by CD8⁺ CD11aʰⁱ T cells after re-stimulation. (B) Representative FACS plots of CD8⁺ CD11aʰⁱ T cells positive for intracellular IFNγ expression after re-stimulation with the SSP2/TRAP₁₃₀₋₁₃₈ peptide. Numbers show the percentage of IFNγ produced by CD8⁺ CD11aʰⁱ T cells.

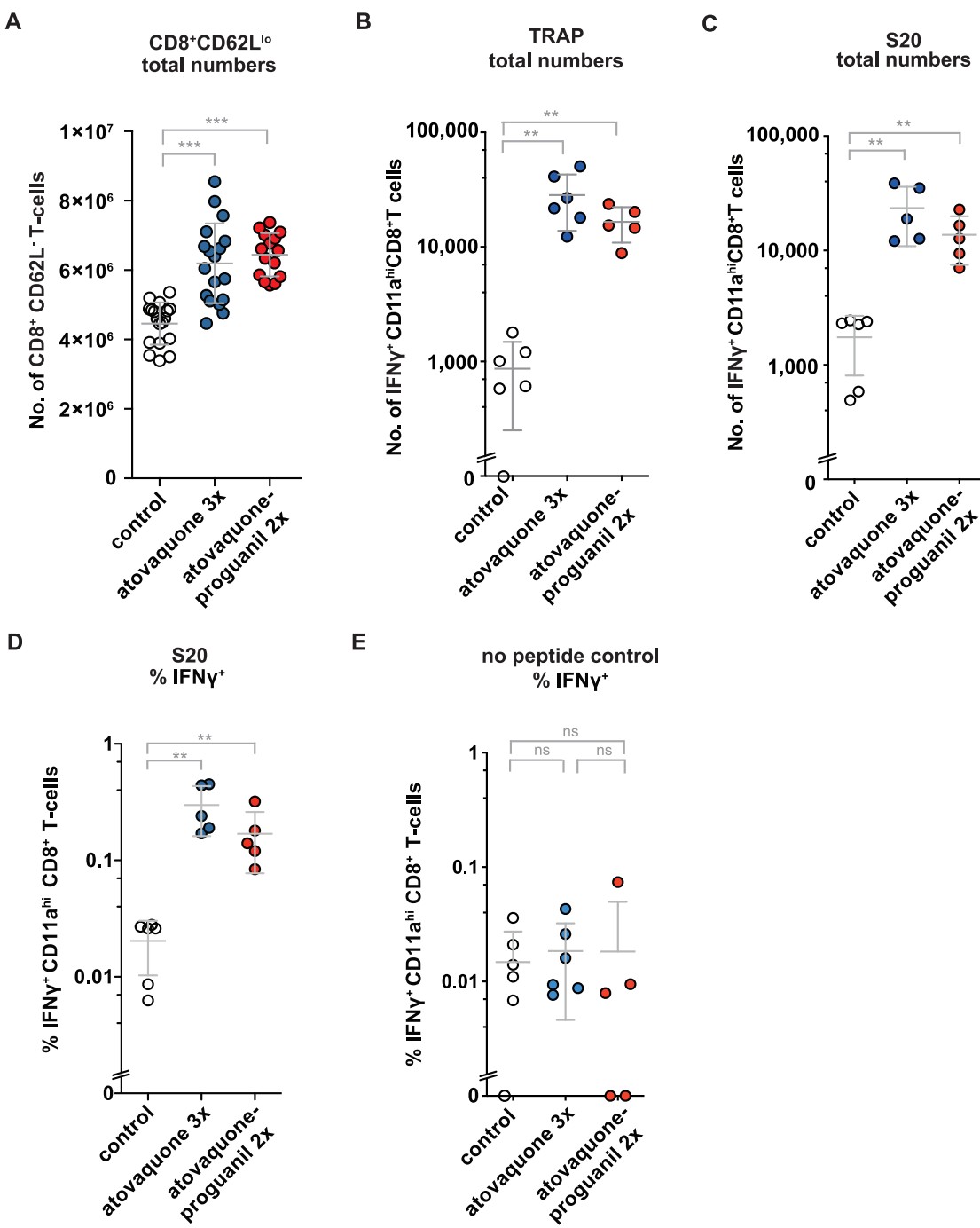

◀ **Figure EV4. Effector memory T cells and proportion of SSP2/TRAP$_{130-138}$- and S20$_{318-326}$-experienced CD8$^+$ CD11a$^{hi}$ T cells.**

Mice were immunised by co-administration of sporozoites and a single dose of atovaquone (3 mg/kg i.p.) or atovaquone–proguanil (3/1.2 mg/kg i.p.). Mice were immunised three times (atovaquone co-administration) and twice (atovaquone–proguanil co-administration). Naive mice served as controls. Sporozoite challenge was done by i.v. injection of 10$^4$ sporozoites 3–4 weeks after the last immunisation. (A) Quantification of total CD8$^+$ CD62L$^{lo}$ T cells from spleens of immunised or control mice. Shown are mean values (±S.D.). Control vs. atovaquone, ***$p < 0.0001$; control vs. atovaquone–proguanil, ***$p < 0.0001$; Mann–Whitney $U$-test. Colour code: white circles, control ($n = 6$); blue circles, atovaquone ($n = 6$); red circles, atovaquone–proguanil ($n = 5$). Cells were quantified in three technical replicates. (B) Quantification of numbers of SSP2/TRAP$_{130-138}$ peptide-specific IFNγ-secretion by CD8$^+$ CD11a$^{hi}$ T cells from spleens of immunised or control mice. Shown are mean values (±S.D.). Control vs. atovaquone, **$p = 0.0022$; control vs. atovaquone–proguanil, **$p = 0.0043$; Mann–Whitney $U$-test. Colour code: white circles, control ($n = 6$); blue circles, atovaquone ($n = 6$); red circles, atovaquone–proguanil ($n = 5$). (C, D) Quantification of numbers (C) and percentage (D) of S20$_{318-326}$ peptide-specific IFNγ-secretion by CD8$^+$ CD11a$^{hi}$ T cells from the spleens of immunised or control mice. Shown are mean values (±S.D.). (C) Control vs. atovaquone, **$p = 0.0043$; control vs. atovaquone–proguanil, **$p = 0.0043$. (D) Control vs. atovaquone, **$p = 0.0022$; control vs. atovaquone–proguanil, **$p = 0.0043$; Mann–Whitney $U$-test. Colour code: white circles, control ($n = 6$; blue circles, atovaquone ($n = 5$); red circles, atovaquone–proguanil ($n = 5$). (E) Quantification of the percentage of baseline IFNγ-secretion by CD8$^+$ CD11a$^{hi}$ T cells without peptide stimulation. Shown are mean values (±S.D.). ns, non-significant. Control vs. atovaquone, ns $p = 0.6991$; control vs. atovaquone–proguanil, ns $p = 0.53$; Mann–Whitney $U$-test. Colour code: white circles, control ($n = 6$); blue circles, atovaquone ($n = 6$); red circles, atovaquone–proguanil ($n = 5$). Number and nature of the replicates as well as exact $p$ values are shown in Appendix Tables S4 and S5. Source data are available online for this figure.

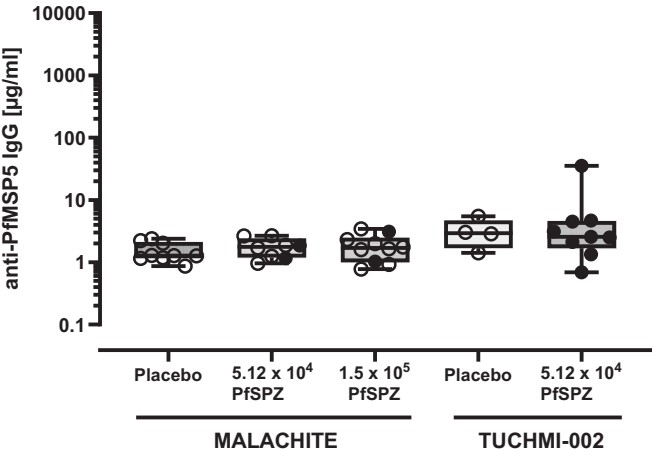

**Figure EV5.   Antibody reactivity against MSP5.**

Reactivity of the plasma samples collected 1 day before CHMI was tested for reactivity against MSP5 in the Malachite and TüCHMI-002 studies. For quantitative analysis, reactivity was estimated in comparison to a coated human IgG standard. Boxplots display median, interquartile range (IQR), and the full range of the data. All ELISA experiments were performed in three technical replicates. No significant differences were found between the groups. Source data are available online for this figure.

