## [Peer Review File · EMBO Molecular Medicine]

Complete Attenuation of *Plasmodium falciparum* Sporozoites by Atovaquone-Proguanil

Steffen Borrmann, Zita Sulyok, Katja Mueller, Rolf Fendel, Mihaly Sulyok, Johannes Friesen, Albert Lalremruata, Thaisa Sandri, The Nguyen, Carlos Calle, Annette Knoblich, Stephanie Sefried, Javier Ibanez, Freia-Raphaella Lorenz, Henri Heimann, David Weller, Regina Steuder, Selorme Adukpo, Patricia Granados Bayon, Zsofia Molnar, Meral Esen, Wolfram Metzger, Eric James, Adam Ruben, Yonas Abebe, Sumana Chakravarty, Anita Manoj, Natasha KC, Tooba Murshedkar, Julius Hafalla, Tamirat Woldearegai, Fiona O'Rourke, Jana Held, Pete Billingsley, B. Sim, Thomas Richie, Stephen Hoffman, Peter Kremsner, Kai Matuschewski, and Benjamin Mordmüller

Corresponding authors: Steffen Borrmann (steffen.borrmann@uni-tuebingen.de) , Benjamin Mordmüller (benjamin.mordmueller@radboudumc.nl)

Review Timeline:

Submission Date:	14th Jan 25
Editorial Decision:	6th Feb 25
Revision Received:	17th Jul 25
Accepted:	22nd Jul 25

Editor: Zeljko Durdevic

Transaction Report:

6th Feb 2025

Dear Prof. Borrmann,

Thank you for the submission of your manuscript to EMBO Molecular Medicine. We have now received feedback from the three reviewers who agreed to evaluate your manuscript. As you will see from their reports pasted below all three referees support publication of your manuscript raising important but minor concerns. Therefore, I am pleased to inform you that we will be able to accept your manuscript pending the following final amendments:

- 1) Please address all referee concerns. No additional experiments are required. Particular attention should be given to streamlining the manuscript presentation.
- 2) Authors: Please make sure that all authors listed in the manuscript are also in our submission system. Currently, Javier Ibáñez, Freia-Raphaella Lorenz and Selorme Adukpo are missing in our submission system.
- 3) Figures: Please upload main figures and EV figures as individual, high-resolution files in TIFF, EPS or PDF format. The legends should be compiled at the end of the manuscript text, with the EV figure legends after the main figure legends and with the heading "Expanded View Figure Legends" Please check "Author Guidelines" for more information:
<https://www.embopress.org/page/journal/17574684/authorguide#figureformat>
<https://www.embopress.org/page/journal/17574684/authorguide#expandedview>
- 4) Author checklist: Please submit a complete checklist. <https://www.embopress.org/pb-assets/embo-site/EMBO%20Press%20Author%20Checklist-1642513524327.xlsx>
- 5) In the main manuscript file, please do the following:
 - Please address all comments suggested by our data editors listed below:
 - o Figure legends:
 1. Please note that the exact p values are not provided in the legends of figures 1C, D; 2B-D; EV1 B, EV4 A-D.
 2. Please indicate the statistical test used for data analysis in the legends of figures 6B, C.
 3. Please note that the box plots need to be defined in terms of minima, maxima in the legend of figure 5A-C
 4. Please note that the box plots need to be defined in terms of minima, maxima, centre, bounds of box and whiskers, and percentile in the legends of figures 6C, EV5
 5. Please note that information related to n is missing in the legends of figures 1C, D; 5A-C; 6B, C; EV1 B, C; EV5
 - Add up to 5 keywords.
 - Figures should be called out in a subsequent order. Currently Fig 5 is called out before Fig 6. Please correct.
 - Remove all figures.
 - In Methods, provide the statement that informed consent was obtained from all human subjects and confirm that the experiments conformed to the principles set out in the WMA Declaration of Helsinki and the Department of Health and Human Services Belmont Report.
 - In Methods, add statistical paragraph that should reflect all information that you have filled in the Authors Checklist, especially regarding randomization, blinding, replication etc.
 - Indicate in legends exact n and exact p values, not a range, along with the statistical test used. To keep the figures "clear" some authors found providing an Appendix table Sx with all exact p-values preferable. You are welcome to do this if you want to.
 - Please include structured Methods section that includes a Reagents and Tools Table (should be uploaded as a separate file) followed by a Methods and Protocols section. More information on how to adhere to this format as well as downloadable templates (.docx) for the Reagents and Tools Table can be found in our author guidelines:
<https://www.embopress.org/page/journal/17574684/authorguide#structuredmethods>
An example of a paper with Structured Methods can be found here:
<https://www.embopress.org/doi/full/10.1038/s44320-024-00037-6#sec-4>
 - Rename "Conflict of interests" to "Disclosure Statement & Competing Interests". We updated our journal's competing interests policy in January 2022 and request authors to consider both actual and perceived competing interests. Please review the policy <https://www.embopress.org/competing-interests> and update your competing interests if necessary.
 - Author contributions: Please remove it from the manuscript and specify author contributions in our submission system. CRediT has replaced the traditional author contributions section because it offers a systematic machine-readable author contributions format that allows for more effective research assessment. You are encouraged to use the free text boxes beneath each contributing author's name to add specific details on the author's contribution. More information is available in our guide to authors:
<https://www.embopress.org/page/journal/17574684/authorguide#authorshippinguidelines>
 - Please rename "Data Availability Section" to "Data Availability" and replace current text with the sentence "This study includes no data deposited in external repositories."
- 6) Funding: Please make sure that information about all sources of funding are complete in both our submission system and in the manuscript. Currently, Boehringer Ingelheim Foundation, the Max Planck Society and AKF grant 346-0-0 is missing in our submission system.
- 7) Appendix: Please move "Safety of immunization with PfSPZ-CVac with AP chemoprophylaxis" to Methods. The study flow chart should be named "Appendix Figure S1" with a short legend added and a callout in the main manuscript text. The tables should be renamed Appendix Table S1, Appendix Table S2 and Appendix Table S3 and called out in the main manuscript text.

On the title page add table of content with page numbers.

8) Synopsis:

- Synopsis text: Please remove the it from the main manuscript text and upload it as a separate .doc file.
- Synopsis image: Please provide a visual abstract as a high-resolution jpeg file 550 pixels wide x 200-600 pixels high to illustrate your article.
- Please check your synopsis text and image before submission with your revised manuscript. Please be aware that in the proof stage minor corrections only are allowed (e.g., typos).

9) As part of the EMBO Publications transparent editorial process initiative (see our Editorial at <http://embomolmed.embopress.org/content/2/9/329>), EMBO Molecular Medicine will publish online a Review Process File (RPF) to accompany accepted manuscripts. This file will be published in conjunction with your paper and will include the anonymous referee reports, your point-by-point response and all pertinent correspondence relating to the manuscript. Let us know whether you agree with the publication of the RPF and as here, if you want to remove or not any figures from it prior to publication. Please note that the Authors checklist will be published at the end of the RPF.

10) Please provide a point-by-point letter INCLUDING my comments as well as the reviewer's reports and your detailed responses (as Word file).

I look forward to reading a new revised version of your manuscript as soon as possible.

Yours sincerely,

Zeljko Durdevic

*** Instructions to submit your revised manuscript ***

- 1) a .docx formatted version of the manuscript text (including Figure legends and tables)
- 2) Separate figure files*
- 3) supplemental information as Expanded View and/or Appendix. Please carefully check the authors guidelines for formatting Expanded view and Appendix figures and tables at <https://www.embopress.org/page/journal/17574684/authorguide#expandedview>
- 4) a letter INCLUDING the reviewer's reports and your detailed responses to their comments (as Word file).
- 5) The paper explained: EMBO Molecular Medicine articles are accompanied by a summary of the articles to emphasize the major findings in the paper and their medical implications for the non-specialist reader. Please provide a draft summary of your article highlighting
 - the medical issue you are addressing,
 - the results obtained and

- their clinical impact.

This may be edited to ensure that readers understand the significance and context of the research.

Please refer to any of our published articles for an example.

6) Author contributions: the contribution of every author must be detailed in a separate section.

7) EMBO Molecular Medicine now requires a complete author checklist

(<https://www.embopress.org/page/journal/17574684/authorguide>) to be submitted with all revised manuscripts. Please use the checklist as guideline for the sort of information we need WITHIN the manuscript. The checklist should only be filled with page numbers where the information can be found. This is particularly important for animal reporting, antibody dilutions (missing) and exact values and n that should be indicated instead of a range.

8) Every published paper now includes a 'Synopsis' to further enhance discoverability. Synopses are displayed on the journal webpage and are freely accessible to all readers. They include a short stand first (maximum of 300 characters, including space) as well as 2-5 one sentence bullet points that summarise the paper. Please write the bullet points to summarise the key NEW findings. They should be designed to be complementary to the abstract - i.e. not repeat the same text. We encourage inclusion of key acronyms and quantitative information (maximum of 30 words / bullet point). Please use the passive voice. Please attach these in a separate file or send them by email, we will incorporate them accordingly.

You are also welcome to suggest a striking image or visual abstract to illustrate your article. If you do please provide a jpeg file 550 px-wide x 300-600px high.

9) A Conflict of Interest statement should be provided in the main text

10) Please note that we now mandate that all corresponding authors list an ORCID digital identifier. This takes <90 seconds to complete. We encourage all authors to supply an ORCID identifier, which will be linked to their name for unambiguous name identification.

Currently, our records indicate that the ORCID for your account is 0000-0001-9189-4393.

Link Not Available

11) Include a Reagents and Tools Table as part of the Methods section, which can be downloaded from our author guidelines (<https://www.embopress.org/page/journal/17574684/authorguide#structuredmethods>)

Photos 400-800 DPI

*Additional important information regarding figures and illustrations can be found at

<https://bit.ly/EMBOPressFigurePreparationGuideline>. See also figure legend preparation guidelines:

<https://www.embopress.org/page/journal/17574684/authorguide#figureformat>

***** Reviewer's comments *****

Referee #1 (Remarks for Author):

Borrmann et al. describe live sporozoite immunization with chemoprophylaxis using atovaquone-proguanil (AP) in mice and in humans, and compare the efficacy and the immune responses elicited by this protocol with those of immunization by chemoprophylaxis with chloroquine (CQ). Both the pre-clinical and the clinical study are well conducted and technically sound, and the conclusions are generally supported by the data. Although the results are interesting and appropriately discussed, a few concerns should be addressed before publication.

Title/Abstract

The main message the authors chose to convey in the Abstract appears to be that "The complete arrest of high numbers of Pf

sporozoites by single-dose AP should allow a significant dose-frequency reduction of the current daily AP malaria chemoprophylaxis regimen". Conversely, the title uses the term "attenuation" to describe the effect of AP on the parasite's liver stage development, suggesting that the main focus of the manuscript is parasite attenuation for immunization by chemoprophylaxis with sporozoites (CPS). Although the two things are not necessarily contradictory, they do highlight different implications of the authors' findings. In the authors' opinion, is the take-home message of the paper that parasite attenuation by AP constitutes a valid alternative to CPS with CQ, or that malaria chemoprophylaxis regimens by AP currently employed in the field should be revised? Or both? Section "The Paper Explained" appears to suggest that the latter is indeed the main message of the paper, in which case the authors should consider changing the title and the running title, whose use of the term "attenuation" strongly evokes the former. It is only in the Discussion that it becomes clear that the authors consider that they present "two major findings regarding PfSPZ vaccines, and one finding regarding the chemoprophylactic efficacy of a single dose of AP", but they seem to struggle with where to place their emphasis throughout the manuscript.

Introduction

Statement "Both vaccines [RTS,S and R21], however, do not meet a previously defined vaccine efficacy (VE) of at least 75% for {greater than or equal to}2 years (Moorthy et al, 2013)" needs to be toned down. A study in LID, titled "Efficacy and immunogenicity of R21/Matrix-M vaccine against clinical malaria after 2 years' follow-up in children in Burkina Faso: a phase 1/2b randomised controlled trial" (PMID 36087586) has shown up to 80% efficacy of R21/MM after 2 years, following the booster vaccination.

The Introduction is very imbalanced in terms of how much it addresses whole-sporozoite vaccines vs AP chemoprophylaxis. This brings back the issue of the main messages of the paper, and what the authors choose to emphasize about their findings. A lot is described about vaccination with metabolically active sporozoites, and then, only at the very end of the Introduction, are some vague notions about the mode of action of atovaquone and proguanil, and a brief mention of parasite resistance to atovaquone, provided. Conversely, far from enough information is given concerning the mode of use of AP for malaria chemoprophylaxis, which appears to be one of the main implications of the study. The Introduction needs to be more balanced and include the most relevant information to explain the current context of AP chemoprophylaxis.

The main observations/conclusions of a study should be briefly summarized at the end of the Introduction. This is absent and should, therefore, be corrected.

Results

- Fig. 3 and Fig. 4 are called in the text at the same time, at the end of the paragraph describing the outcome of CHMI in groups A and B. These two figures should be combined as panels of one figure, as they present complementary information.

Discussion

The authors should discuss the implications of their findings in the wider context of whole-sporozoite vaccination against malaria, most notably the recently reported PflARC2 vaccine candidate.

When discussing early- vs late-arresting liver parasites, the authors should mention and cite the recent work by Lamers et al (PMID: 39565990).

Seen as one of the study's main findings regards the chemoprophylactic efficacy of a single dose of AP, a thorough discussion about AP doses and regimens currently employed for chemoprophylaxis in the field, and additional details on how the authors propose that these should be revised is necessary.

Referee #2 (Comments on Novelty/Model System for Author):

Borrmann et al test the efficacy of the antimalarial atovaquone and atovaquone/proguanil in combination for use as a chemo-attenuation drug to improve liver stage vaccination strategies with live parasites. The study found promising data in rodent models, but this did not translate to useful protective efficacy in controlled human malaria models compared to the use of chloroquine for chemical attenuation. Antibodies to two liver stage and one merozoite stage antigen were found to be lower for atovaquone/parasite vaccinated individuals than seen for the more protective chloroquine/parasite vaccinated individuals, a potentially useful observation when trying to identify the most protective antigens. The study identified that a higher than commonly used dose given just once was able to protect from blood stage infection, suggesting that an increased drug dose could enable effective prophylaxis with a single dose. Confirmation that the three antigens identified could be associated with improved vaccine protection and that atovaquone based treatment regimens could be reduced from 4 to a single dose and offer protection from infection would both need further confirmation in future studies. The work is overall well done, but some of the data presentation could be improved.

Referee #2 (Remarks for Author):

Borrmann et al test the efficacy of the antimalarial atovaquone and atovaquone/proguanil in combination for use as a chemo-attenuation drug to improve liver stage vaccination strategies with live parasites.

Major comments

- The number of biological replicates for in vitro experiments is not provided throughout the manuscript. If these experiments were done just the once, then that is not ideal given that some are just simple ELISAs and repeat experiments would strengthen reproducibility. It would be good to have repeats. If the authors have a reason this can't be done then the number of biological replicates should be stated so the reader can assess that as well.
- Why was no statistical comparison done for the data in Figure 5 and EV5? It is done elsewhere in the manuscript.
- The association with MSP5 and protection in the chloroquine model is not discussed in the discussion at all. Likewise, the antigens that had antibody levels higher in the atovaquone treatment are also not mentioned. These should be discussed.
- Figure sizing (e.g. Fig 5, but others as well) are all different and the text is difficult to read for some. Suggest standardizing the Figures better.
- Fig 5E: Suggest moving this description of the treatment plan to the top of the figure, before the resulting data is presented.

Minor comments

- Page 8 row 8 - Mueller et al, 2005 (Remove extra b typo)
- Page 8 row 14 - Vaccine efficacy (VE) shortform should be introduced here instead of Page 11 row 13
- Page 10 row 4 - Suspect that the figure is meant to be EV1b
- Page 10 row 14 - What is the independent experiment?
- Page 16 row 1. Instead of 'a few' could a number be given? The number of samples which don't show any change is given.
- Page 25 last row - GraphPad Prism 5 is mentioned on page 46, Graph-Pad Prism v9.4 is mentioned in the methods.
- Page 45 row 12 - Open circles in are not visible in Figure 5
- On page 46 the word 'verum' is used a couple of times. The definitions I could find for this word does not really fit the context its being used for. I assume this is a typo (probably meant to be serum). However, if not, suggest using simpler terminology.
- Page 48 Figure 2B - What was the reason for having the atovaquone 2X data separate? Could this be explained?
- Page 50 Figure 4 - Is there a reason the data only goes up to 12.5 days post inoculation?
- Page 52 Figure 6C - Based on how data have been previously presented, would it be worth changing the order of the boxplots data to violet, red and blue
- Appendix - Tables have no number?
- Page 56 - 'exemplary' or 'examples'.

Referee #3 (Comments on Novelty/Model System for Author):

Review of Bormann et al.

This is an important piece of work, exploring the concept of whole parasite vaccination with drug coverage, extending previous studies using chemo-vaccination (CVac) by co-administration of blood-stage targeting Chloroquine (CQ) alongside sporozoite immunization (PfSPZ) but with an alternative drug/drug combination (Atovaquone/Proguanil, AP). With stagnating rates of malaria incidence/death - work on malaria vaccines is as important today as it has ever been & therefore of keen medical importance.

The study is thorough and well-executed, it is well controlled. The model (mouse) is the gold standard in the field for pre-clinical testing in humans, and is used accordingly. The move to humans is welcome as it extends and clarifies (though not always agreeing) with observations in the model. This is important to publish.

The study clearly shows some novel properties for the PfSPZ-CVac (AP), noting its ability to confer complete attenuation of parasites (hence the title) without blood stage breakthrough (from mouse to human) but its poor efficacy as a PfSPZ-CVac strategy, especially in comparison to CQ.

Referee #3 (Remarks for Author):

Review of Bormann et al.

This is an important piece of work, exploring the concept of whole parasite vaccination with drug coverage, extending previous studies using chemo-vaccination (CVac) by co-administration of blood-stage targeting Chloroquine (CQ) alongside sporozoite immunization (PfSPZ) but with an alternative drug/drug combination (Atovaquone/Proguanil, AP). With stagnating rates of

malaria incidence/death - work on malaria vaccines is as important today as it has ever been & therefore of keen medical importance.

The study is thorough and well-executed, it is well controlled. The model (mouse) is the gold standard in the field for pre-clinical testing in humans, and is used accordingly. The move to humans is welcome as it extends and clarifies (though not always agreeing) with observations in the model. This is important to publish.

The study clearly shows some novel properties for the PfSPZ-CVac (AP), noting its ability to confer complete attenuation of parasites (hence the title) without blood stage breakthrough (from mouse to human) but its poor efficacy as a PfSPZ-CVac strategy, especially in comparison to CQ.

As a vaccination strategy, the study represents a bit of a dead-end of the AP idea (for PfSPZ co-administration), however, by studying the immune responses comparing the two different approaches (AP vs CQ) it elucidates antibody correlates of protection (three antigens in particular, LISP, LSA2 and MSP5) and adds further weight to the growing awareness around the importance of cellular immune responses, and their reliance on later stages of intra-hepatic parasite development, to any future vaccine efficacy. These (and other subtle observations) are important.

A few comments below suggest some text clarifications/additions that would aid the communication of the study's findings, but do not require further experimental work. Please note, that whilst this is a fresh submission, I have reviewed a previous version of this manuscript and believe it is markedly improved from that version (I supported its acceptance/publication then, and do so now):

1. I am still uncertain whether the title fully conveys the message of the paper. It certainly points to the ability of AP to attenuate parasites pre-blood stages. However, wouldn't it be good to shout about it's role in uncovering novel serological correlates of protection?

2. The paper explained and synopsis/abstract all use different language/phrases to communicate the impact of the work. It'd be good to harmonise these e.g. around the phrase: "a signature of adaptive immunity against mature liver stages" or "a biomarker for cryptic liver stage expansion" "association of antibodies against proteins X, Y, and Z". Each of these is not the same. My feeling is the latter (in the abstract) is likely the most accurate.

3. The lower levels of anti-CSP (given everything else is equal) is quite intriguing. I accept the idea that further expression of the protein through intra-hepatic stages might boost the response (although parasites are intracellular). I think it may also be worth exploring immunosuppressive effects of intra-hepatic development (in early stages, released later on) that might be at play - though much harder to define.

4. I think it's worth clearly stating in the results (e.g. p15 penultimate paragraph) that the anti-PfCSP antibody independent mechanisms at play are likely cellular immunity i.e. T-cells.

5. I am trying to get my head around why antibodies against LISP2 and LSA1 appear - as (perhaps naively) I'd think the proteins would be less of an antibody target if they are from the intra-hepatic schizont. MSP5 I get as it will get exposed on the merozoite surface. Is it well known/characterised that liver stage (intra-hepatic) parasite proteins are proven to be the targets of antibodies? Naively, I'd think they'd be hidden. Or are these proteins on surface of blood stages too?

6. Whilst recognising that no paper is ever up to date with the pace of publishing, I think it'd be important to give a nod to the work in NEJM: <https://www.nejm.org/doi/full/10.1056/NEJMoa2313892> concerning the GA2 protective efficacy in clinical trials. It's findings clearly align with many of the observations here regarding cellular immunity.

******* Reviewer's comments *******

We are very grateful for the quick and helpful comments by the three reviewers.

Referee #1 (Remarks for Author):

Borrmann et al. describe live sporozoite immunization with chemoprophylaxis using atovaquone-proguanil (AP) in mice and in humans, and compare the efficacy and the immune responses elicited by this protocol with those of immunization by chemoprophylaxis with chloroquine (CQ). Both the pre-clinical and the clinical study are well conducted and technically sound, and the conclusions are generally supported by the data. Although the results are interesting and appropriately discussed, a few concerns should be addressed before publication.

We are thankful for these are very encouraging comments.

Title/Abstract

The main message the authors chose to convey in the Abstract appears to be that "The complete arrest of high numbers of Pf sporozoites by single-dose AP should allow a significant dose-frequency reduction of the current daily AP malaria chemoprophylaxis regimen". Conversely, the title uses the term "attenuation" to describe the effect of AP on the parasite's liver stage development, suggesting that the main focus of the manuscript is parasite attenuation for immunization by chemoprophylaxis with sporozoites (CPS). Although the two things are not necessarily contradictory, they do highlight different implications of the authors' findings. In the authors' opinion, is the take-home message of the paper that parasite attenuation by AP constitutes a valid alternative to CPS with CQ, or that malaria chemoprophylaxis regimens by AP currently employed in the field should be revised? Or both? Section "The Paper Explained" appears to suggest that the latter is indeed the main message of the paper, in which case the authors should consider changing the title and the running title, whose use of the term "attenuation" strongly evokes the former. It is only in the Discussion that it becomes clear that the authors consider that they present "two major findings regarding PfSPZ vaccines, and one finding regarding the chemoprophylactic efficacy of a single dose of AP", but they seem to struggle with where to place their emphasis throughout the manuscript.

We agree that the main messages were not consistently explained. We have rewritten the Impact paragraph of The Paper Explained, which is now reflecting the three major findings in the concluding remarks in the Discussion. However, due to the required brevity (stringent word count limit of the abstract) and because we had previously received the recommendation to select a more succinct title, we would like to abstain from incorporating all major findings in both title and abstract.

Introduction

Statement "Both vaccines [RTS,S and R21], however, do not meet a previously defined vaccine efficacy (VE) of at least 75% for {greater than or equal to}2 years (Moorthy et al, 2013)" needs to be toned down. A study in LID, titled "Efficacy and immunogenicity of R21/Matrix-M vaccine against clinical malaria after 2 years' follow-up in children in Burkina Faso: a phase 1/2b randomised controlled trial" (PMID 36087586) has shown up to 80% efficacy of R21/MM after 2 years, following the booster vaccination.

We agree and have better characterised our statement by adding "... in phase 3 trials." This is because the, nonetheless promising, 80% (95% CI, 72 to 85) vaccine efficacy at 2 years following a booster in the high-dose adjuvant group is limited to relatively small numbers (n=137 in this group) and in only at a single study site with pronounced seasonality, which is not representative for most of malaria-endemic African countries. No data have been published on the two-years performance of R21/Matrix-M in the subsequent phase 3 trial.

The Introduction is very imbalanced in terms of how much it addresses whole-sporozoite vaccines vs AP chemoprophylaxis. This brings back the issue of the main messages of the paper, and what the authors choose to emphasize about their findings. A lot is described about vaccination with metabolically active sporozoites, and then, only at the very end of the Introduction, are some vague notions about the mode of action of atovaquone and proguanil, and a brief mention of parasite resistance to atovaquone, provided. Conversely, far from enough information is given concerning the mode of use of AP for malaria chemoprophylaxis, which appears to be one of the main implications of the study. The Introduction needs to be more balanced and include the most relevant information to explain the current context of AP chemoprophylaxis.

We agree and have thus rewritten and significantly expanded the second last paragraph of the Introduction dealing with AP chemoprophylaxis, including a discussion of the rationale for the selected dose.

The main observations/conclusions of a study should be briefly summarized at the end of the Introduction. This is absent and should, therefore, be corrected.

We understand the intention. However, we would like to leave it as is since the main findings are being summarised already repeatedly (now also including in the Paper Explained section). However, we are open to add such a summary should it be also desired by the editor/editorial team at EMBO Molecular Medicine.

Results

- Fig. 3 and Fig. 4 are called in the text at the same time, at the end of the paragraph describing the outcome of CHMI in groups A and B. These two figures should be combined as panels of one figure, as they present complementary information.

We agree and have thus combined Fig. 3 and Fig. 4 plus updated the manuscript throughout accordingly (e.g., previous Fig. 5 is now Fig. 4 etc).

Discussion

The authors should discuss the implications of their findings in the wider context of whole-sporozoite vaccination against malaria, most notably the recently reported PfLARC2 vaccine candidate.

We fully agree, this has been added as a concise penultimate paragraph in the Discussion.

When discussing early- vs late-arresting liver parasites, the authors should mention and cite the recent work by Lamers et al (PMID: 39565990).

This has been incorporated in the above mentioned penultimate paragraph in the Discussion, including a reference to a separate spotlight article on this work by some of us.

Seen as one of the study's main findings regards the chemoprophylactic efficacy of a single dose of AP, a thorough discussion about AP doses and regimens currently employed for chemoprophylaxis in the field, and additional details on how the authors propose that these should be revised is necessary.

A discussion of the possible way forward is already included in the Paper Explained section at a relatively fine-grained detail level (as far as the existing evidence permits at this stage).

Referee #2 (Comments on Novelty/Model System for Author):

Borrmann et al test the efficacy of the antimalarial atovaquone and atovaquone/proguanil in combination for use as a chemo-attenuation drug to improve liver stage vaccination strategies with live parasites. The study found promising data in rodent models, but this did not translate to useful protective efficacy in controlled human malaria models compared to the use of chloroquine for chemical attenuation. Antibodies to two liver stage and one merozoite stage antigen were found to be lower for atovaquone/parasite vaccinated individuals than seen for the more protective chloroquine/parasite vaccinated individuals, a potentially useful observation when trying to identify the most protective antigens. The study identified that a higher than commonly used dose given just once was able to protect from blood stage infection, suggesting that an increased drug dose could enable effective prophylaxis with a single dose. Confirmation that the three antigens identified could be associated with improved vaccine protection and that atovaquone based treatment regimens could be reduced from 4 to a single dose and offer protection from infection would both need further confirmation in future studies. The work is overall well done, but some of the data presentation could be improved.

We're grateful for this encouraging assessment of our work.

Referee #2 (Remarks for Author):

Borrmann et al test the efficacy of the antimalarial atovaquone and atovaquone/proguanil in combination for use as a chemo-attenuation drug to improve liver stage vaccination strategies with live parasites.

Major comments

-The number of biological replicates for in vitro experiments is not provided throughout the manuscript. If these experiments were done just the once, then that is not ideal given that some are just simple ELISAs and repeat experiments would strengthen reproducibility. It would be good to have repeats. If the authors have a

reason this can't be done then the number of biological replicates should be stated so the reader can assess that as well.

We have clarified that these pre-clinical *in vitro* experiments were done only once (as in previous work by us and others). This concerns the data presented in Figure 1A, EV1A-C, and Fig. 2D. Concerning the human immunology data, however, ELISAs were done in three technical replicates. We have updated the respective figure legends with this information.

-Why was no statistical comparison done for the data in Figure 5 and EV5? It is done elsewhere in the manuscript.

We have added the P values for all comparisons in Fig. 5 (now Fig. 4) and added a note in the legend for Fig. EV5 (“no significant differences were found between the groups”).

-The association with MSP5 and protection in the chloroquine model is not discussed in the discussion at all. Likewise, the antigens that had antibody levels higher in the atovaquone treatment are also not mentioned. These should be discussed.

Thanks for pointing out this inconsistency. We have added a discussion on MSP5 in the Discussion: “Using ELISA, we detected higher levels of IgM, but not IgG, antibodies recognizing MSP5 after vaccination with PfSPZ-CVac (CQ) as compared to PfSPZ-CVac (AP). This is reminiscent of previous observations with early arresting gamma-irradiated PfSPZ vaccination (PfSPZ Vaccine), which also elicited a highly focused IgM antibody response to MSP5 in protected vaccinees (Tumbo *et al*, 2024).”

However, we believe that a discussion of antibody responses that were higher in the atovaquone group is not as relevant since these antibody responses clearly didn't contribute to protection (and also vis-à-vis an already expansive Discussion section).

-Figure sizing (e.g. Fig 5, but others as well) are all different and the text is difficult to read for some. Suggest standardizing the Figures better.

Thanks. We can see this point – however, this mainly depends on the final layout in the published article. All figures are provided as vectored graphics and thus can be adjusted freely without any loss to their quality (with the typical limits applying to embedded, pixel-based micrographs). Likewise, if needed fonts could be adjusted separately using standard vector design software (such as Adobe Illustrator). We are prepared, including at short notice, to assist with this process if deemed necessary and/or desirable.

-Fig 5E: Suggest moving this description of the treatment plan to the top of the figure, before the resulting data is presented.

We assume that the referee was referring to Fig. 2E (not 5E). This is also an option but we would like to keep it as is but remain open to further instructions from the editor/editorial team.

Minor comments

-Page 8 row 8 - Mueller et al, 2005 (Remove extra b typo)

This is not a typo but required to distinguish between the two Mueller et al. 2005 papers cited in the manuscript.

-Page 8 row 14 - Vaccine efficacy (VE) shortform should be introduced here instead of Page 11 row 13

Thanks. In fact, it had been introduced even before page 11 (on page 6 in the 2nd paragraph of the Introduction). However, throughout the manuscript we have used “vaccine efficacy” and “VE” not consistently (and even, re-introduced the abbreviation). Hence, we have used VE across the whole manuscript after the initial introduction on page 6.

-Page 10 row 4 - Suspect that the figure is meant to be EV1b

It is actually referencing all panels (A-C) in Fig. EV1.

-Page 10 row 14 - What is the independent experiment?

Thanks, this was clarified by adding “In this experiment, AP...” to the next sentence (which explains the “independent experiment” mentioned in the previous sentence).

-Page 16 row 1. Instead of 'a few' could a number be given? The number of samples which don't show any change is given.

Thanks, we have now stated the actual number (3).

-Page 25 last row - GraphPad Prism 5 is mentioned on page 46, Graph-Pad Prism v9.4 is mentioned in the methods.

Thanks, this has been corrected (version v. 10.41 was correct).

-Page 45 row 12 - Open circles in are not visible in Figure 5

Thanks, this has been corrected in the legend of Fig. 4 (formerly, Fig. 5).

-On page 46 the word 'verum' is used a couple of times. The definitions I could find for this word does not really fit the context its being used for. I assume this is a typo (probably meant to be serum). However, if not, suggest using simpler terminology.

It is actually explained in the previous figure legend (for Fig. 3): “...Group A (verum) and B (placebo)...”. Thus, “verum donors” refers to donors who received an intervention (immunisation with live sporozoites attenuated by concomitant administration of an antimalarial drug) and not placebo. Verum, if a bit old school, remains a commonly used term in clinical trials.

-Page 48 Figure 2B - What was the reason for having the atovaquone 2X data separate? Could this be explained?

These were two separate experiments, including with separate controls and since qPCR data can differ slightly from experiment to experiment we decided to show them separately. However, for Fig. 2A we believe that the survival data from the different can be usefully combined.

-Page 50 Figure 4 - Is there a reason the data only goes up to 12.5 days post inoculation?

Thanks, this has been corrected (now Fig. 3B due to the consolidation of former Fig. 3 and Fig. 4). The x axis is now correctly labelled with ticks at 2-day intervals (rather than 2.5-day intervals as before) in line with the once-daily follow-up.

-Page 52 Figure 6C - Based on how data have been previously presented, would it be worth changing the order of the boxplots data to violet, red and blue

Thanks, we have corrected the figure legend to reflect the order of the presentation of the data in Fig. 5C (formerly, Fig. 6C).

-Appendix - Tables have no number?

No, we hope to have used the EMBO Molecular Medicine guidelines for authors correctly (but remain open to further editorial advice).

-Page 56 - 'exemplary' or 'examples'.

Thanks. We had used “exemplary” to indicate “representative” and to improve clarity we have replaced “exemplary” with “representative”.

Referee #3 (Comments on Novelty/Model System for Author):

Review of Bormann et al.

This is an important piece of work, exploring the concept of whole parasite vaccination with drug coverage, extending previous studies using chemo-vaccination (CVac) by co-administration of blood-stage targeting Chloroquine (CQ) alongside sporozoite immunization (PfSPZ) but with an alternative drug/drug combination (Atovoquone/Proguanil, AP). With stagnating rates of malaria incidence/death - work on malaria vaccines is as important today as it has ever been & therefore of keen medical importance.

The study is thorough and well-executed, it is well controlled. The model (mouse) is the gold standard in the field for pre-clinical testing in humans, and is used accordingly. The move to humans is welcome as it extends and clarifies (though not always agreeing) with observations in the model. This is important to publish.

The study clearly shows some novel properties for the PfSPZ-CVac (AP), noting its ability to confer complete attenuation of parasites (hence the title) without blood stage breakthrough (from mouse to human) but its poor efficacy as a PfSPZ-CVac strategy, especially in comparison to CQ.

Referee #3 (Remarks for Author):

Review of Bormann et al.

This is an important piece of work, exploring the concept of whole parasite vaccination with drug coverage, extending previous studies using chemo-vaccination (CVac) by co-administration of blood-stage targeting Chloroquine (CQ) alongside sporozoite immunization (PfSPZ) but with an alternative drug/drug combination (Atovoquone/Proguanil, AP). With stagnating rates of malaria incidence/death - work on malaria vaccines is as important today as it has ever been & therefore of keen medical importance.

The study is thorough and well-executed, it is well controlled. The model (mouse) is the gold standard in the field for pre-clinical testing in humans, and is used accordingly. The move to humans is welcome as it extends and clarifies (though not always agreeing) with observations in the model. This is important to publish.

We're delighted to read these compliments.

The study clearly shows some novel properties for the PfSPZ-CVac (AP), noting its ability to confer complete attenuation of parasites (hence the title) without blood stage breakthrough (from mouse to human) but its poor efficacy as a PfSPZ-CVac strategy, especially in comparison to CQ.

We agree.

As a vaccination strategy, the study represents a bit of a dead-end of the AP idea (for PfSPZ co-administration), however, by studying the immune responses comparing the two different approaches (AP vs CQ) it elucidates antibody correlates of protection (three antigens in particular, LISP, LSA2 and MSP5) and adds further weight to the growing awareness around the importance of cellular immune responses, and their reliance on later stages of intra-hepatic parasite development, to any future vaccine efficacy. These (and other subtle observations) are important.

Again, many thanks for this encouraging assessment of our work.

A few comments below suggest some text clarifications/additions that would aid the communication of the study's findings, but do not require further experimental work. Please note, that whilst this is a fresh submission, I have reviewed a previous version of this manuscript and believe it is markedly improved from that version (I supported its acceptance/publication then, and do so now):

We are grateful for this reviewer's consistent and patient advice on our manuscript.

- 1. I am still uncertain whether the title fully conveys the message of the paper. It certainly points to the ability of AP to attenuate parasites pre-blood stages. However, wouldn't it be good to shout about it's role in uncovering novel serological correlates of protection?*

We have also internally discussed this topic at great length but came to the conclusion to highlight a main single finding, which believe will have the greatest impact on potentially changing practice of medicine (in this case, potentially a significant reduction of the chemoprophylactic regimen with AP). We hope that this is acceptable to the editorial team?

2. *The paper explained and synopsis/abstract all use different language/phrases to communicate the impact of the work. It'd be good to harmonise these e.g. around the phrase: "a signature of adaptive immunity against mature liver stages" or "a biomarker for cryptic liver stage expansion" "association of antibodies against proteins X, Y, and Z". Each of these is not the same. My feeling is the latter (in the abstract) is likely the most accurate.*

Thanks, we agree. We have thus (also based on the other reviewer's comments) attempted to unify these seemingly mixed messages in the final chapter of the Discussion: "These findings support the contention that the full protection of PfSPZ-CVac (CQ) vs the minimal protection of PfSPZ-CVac (AP) is related to immune responses against antigens expressed during the cryptic liver stage expansion of an infection and that specific antibody responses could be useful biomarkers in future clinical trials of PfSPZ-CVac."

3. *The lower levels of anti-CSP (given everything else is equal) is quite intriguing. I accept the idea that further expression of the protein through intra-hepatic stages might boost the response (although parasites are intracellular). I think it may also be worth exploring immunosuppressive effects of intra-hepatic development (in early stages, released later on) that might be at play - though much harder to define.*

We assume that the reviewer is referring to the ELISA data shown in Fig. 4A,B. We agree that the lower levels of anti-CSP IgG (and IgM) after live sporozoite immunisation attenuated by AP compared to CQ at exactly the same PfSPZ doses remains difficult to explain. At this stage, beyond the possibility that this could be due to continued expression of CSP in maturing liver stages with CQ but not AP, we have no further hypothesis to propose. We feel that potential immunosuppressive effects "of *intra-hepatic development*" is a rather remote possibility (mediated by which mechanism?) and while not impossible, we would not like to speculate on it further in the Discussion. However, with regard to a potential immunosuppressive effect of AP, we had already remarked: "We cannot formally exclude that AP has a heretofore undefined immunosuppressive activity in humans as reported for CQ (Chen *et al*, 2018). We consider this notion unlikely since in over more than 20 years of clinical use of AP no such evidence has been reported."

4. *I think it's worth clearly stating in the results (e.g. p15 penultimate paragraph) that the anti-PfCSP antibody independent mechanisms at play are likely cellular immunity i.e. T-cells.*

Thanks, this has been incorporated.

5. *I am trying to get my head around why antibodies against LISP2 and LSA1 appear - as (perhaps naively) I'd think the proteins would be less of an antibody target if they are from the intra-hepatic schizont). MSP5 I get as it will get exposed on the merozoite surface. Is it well known/characterised that liver stage (intra-hepatic) parasite proteins are proven to be the targets of antibodies? Naively, I'd think they'd be hidden. Or are these proteins on surface of blood stages too?*

This is a very good question that we had also considered but for which we unfortunately don't have a good, definitive answer. We and others definitely do see these antibody responses and as Fabra-García *et al.* have shown (PMID 35167490) some appear to even block intrahepatic development. Neither LISP2 nor LSA1 have been shown to be expressed on blood stage merozoites or infected red blood cells. No change to the manuscript.

6. *Whilst recognising that no paper is ever up to date with the pace of publishing, I think it'd be important to give a nod to the work in NEJM:
<https://www.nejm.org/doi/full/10.1056/NEJMoa2313892> concerning the GA2 protective efficacy in clinical trials. It's findings clearly align with many of the observations here regarding cellular immunity.*

Thanks, this was also pointed out by reviewer #1 and we have added this important recent paper to the reference list.

22nd Jul 2025

Dear Prof. Borrmann,

We are pleased to inform you that your manuscript is accepted for publication and is now being sent to our publisher to be included in the next available issue of EMBO Molecular Medicine.
